# Discovering and Steering Interpretable Concepts in Large Generative Music Models

**Nikhil Singh**[*1]**, Manuel Cherep**[*2]**, Pattie Maes**[2]
[1]Dartmouth College, [2]MIT. * Equal contribution.

musicdiscovery.media.mit.edu

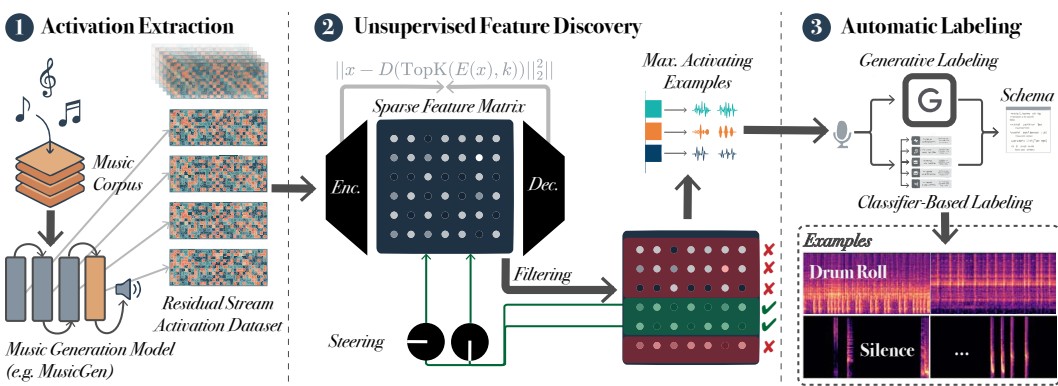

Figure 1: Multi-stage pipeline for discovering and steering interpretable concepts in autoregressive music models. **(1)** Music from a large corpus is passed through a pre-trained generator to extract activations from multiple layers. **(2)** Sparse autoencoders reconstruct activations (also usable for steering), and features are filtered to retain the most viable candidates. **(3)** Retained features are characterized via musical examples and labeled using generative labeling with a multimodal LM and classifier-based labeling with pre-trained models.

## Abstract

The fidelity with which neural networks can now generate content such as music presents a scientific opportunity: these systems appear to have learned implicit theories of such content's structure through statistical learning alone. This offers a potentially new lens on theories of human-generated media. When internal representations align with traditional constructs (e.g. chord progressions in music), they show how such categories can emerge from statistical regularities; when they diverge, they expose limits of existing frameworks and patterns we may have overlooked but that nonetheless carry explanatory power. In this paper, focusing on autoregressive music generators, we introduce a method for discovering interpretable concepts using sparse autoencoders (SAEs), extracting interpretable features from the residual stream of a transformer model. We make this approach scalable and evaluable using automated labeling and validation pipelines. Our results reveal both familiar musical concepts and coherent but uncodified patterns lacking clear counterparts in theory or language. As an extension, we show such concepts can be used to steer model generations. Beyond improving model transparency, our work provides an empirical tool for uncovering organizing principles that have eluded traditional methods of analysis and synthesis.

## 1 Introduction

When humans create, we draw upon a rich vocabulary of concepts. Some of these we can name and explicitly reason about; terms like "symmetry" or "gradient" in the visual arts, for instance. Others shape our media long before we have language for them: consider how the "Hero's Journey" narrative structure infused myths, legends, and epics across cultures for millennia before Joseph Campbell famously theorized it in the 20th century (Campbell, 1949). This gap between practice

and theory reflects a more fundamental pattern: our ability to recognize and use structure often precedes our ability to describe it.

This phenomenon poses an interesting puzzle for machine learning. Deep generative models produce remarkably convincing content through statistical learning, suggesting they learn coherent internal representations of the building blocks of such content. How do we bridge the gap between these models' raw statistical power and the structured conceptual vocabularies humans use? The answer to this could impact whether generative models will remain opaque mimics or become genuine creative collaborators that can engage with and perhaps even extend human creative frameworks.

Consider what it means for a model to "understand" a musical concept like chord progressions. Some neural pattern must respond to and enable their generation, yet this pattern might not correspond neatly to our theoretical description. Connecting our hypotheses to generative models' opaque mechanisms could enable useful new interactions through shared analytical and synthetic vocabularies. Here, music offers a compelling test domain: it combines centuries-long theoretical vocabulary with perceptual patterns resisting verbal description, and exhibits both statistical regularity and cultural variety. It also lacks large-scale data (e.g. paired music-text) that might guide concept discovery. Many efforts have sought to align music generation with human intent, making this a challenging yet useful case for concept discovery in generative models.

This paper introduces a method using sparse autoencoders to extract features from transformer-based music model activations. We demonstrate how these features reveal both familiar patterns aligning with traditional concepts and implicitly learned regularities that elude clear verbal description. In summary, this work makes the following contributions:

1. **A general pipeline for *unsupervised* concept discovery in autoregressive music models.** We show how sparse autoencoders applied to MusicGen residual streams reliably surface interpretable features, extending recent interpretability techniques beyond text and vision into audio/music. This is the first application of SAEs in audio, to our knowledge.

2. **Automated large-scale evaluation of discovered features.** We combine multimodal LLM labeling, pretrained audio classifiers, cross-modal alignment, and other techniques to automatically name and score thousands of latent musical concepts at scale.

3. **Evidence of well-known and subtle musical concepts.** Our method recovers familiar categories (e.g. genres, timbres, instruments) while also surfacing coherent patterns not yet well-described by theoretical constructs, demonstrating that these models may encode musically meaningful structures beyond human-provided labels.

4. **Empirical insight into layer- and model-size effects.** We show how concept interpretability and distinctiveness vary with layer depth, sparsity/expansion settings, and model scale, contributing evidence to the study of feature localization in generative models.

5. **Demonstration of feature steering in generation.** We provide proof-of-concept steering experiments showing that discovered features can be directly manipulated to alter generation outputs, establishing practical utility of the approach for controllable generation.

## 2 RELATED WORK

### 2.1 STATISTICAL LEARNING OF MUSICAL STRUCTURE

Computational approaches to modeling music span rule-based systems, probabilistic models, and modern neural networks. Early Markov models captured local dependencies like chord transitions but failed at longer-range structure (Conklin, 2003), while information-theoretic analyses have revealed statistical bases for harmonic categories (Jacoby et al., 2015). For several years, neural architectures have achieved impressive results in both symbolic and audio modeling (Huang et al., 2018; Van Den Oord et al., 2016; Mehri et al., 2016). More recent systems often model audio directly with transformers or diffusion approaches (Agostinelli et al., 2023; Copet et al., 2024; Liu et al., 2023; Huang et al., 2023). Yet as these models generate increasingly sophisticated music, our understanding of what musical knowledge they acquire lags behind. The progress in generation has coincided with greater opacity of internal structure, motivating our central question: **what do these models actually learn about music?** Our work seeks to make progress on answering this question (see Appendix G for additional music-theoretic context).

## 2.2 Interpretability Through Feature Extraction

Interpreting neural network computations has long been a central goal. Recent work shows that LLM representations often encode concepts in surprisingly localized ways (Templeton et al., 2024), echoing classic sparse coding results where mammalian visual cortex was modeled as learning efficient codes for natural images (Olshausen & Field, 1997). This broader lesson, namely that sparse latent structure can be computationally and biologically advantageous, has motivated the development of interpretability methods. A recent line trains *sparse autoencoders* (SAEs) (Mallat & Zhang, 1993; Templeton et al., 2024) on transformer activations, enforcing sparsity to extract interpretable components. Formally, with encoder $E$ and decoder $D$, one optimizes:

$$\min_{E,D} \mathbb{E}_x[\|x - D(E(x))\|_2^2 + \lambda\|E(x)\|_1] \tag{1}$$

This constraint is thought to encourage discovery of reusable "atomic" concepts rather than memorized reconstructions. Learned features can then be labeled using automated methods (e.g. LLMs applied to maximally activating sequences). Extending such approaches to music generators is nontrivial, given their hierarchical temporal structure and mixed discrete–continuous features. **Our work develops adaptations to apply these principles to the musical domain.**

## 2.3 Interpreting Music Generation Models

Given a vocabulary of known concepts and example data to match, one strategy toward interpreting music generation models is *probing*. While probing has a rich history in NLP, work has recently begun to apply it to music models. Prior work (Wei et al., 2024a) introduced the *SynTheory* dataset to systematically probe music foundation models on core music theory concepts like tempo, intervals, and chords, revealing that certain layers capture these concepts more strongly than others. Similarly, other work investigates chord and pitch representations in MusicGen and find that musical concepts become more evident in deeper layers, yet may not be linearly separable without additional interventions in the hidden space (Ma & Xia, 2024). Beyond these direct probing and manipulation techniques, recent work (Vásquez et al., 2024) uses the *DecoderLens* (Langedijk et al., 2023) method to interpret intermediate activations, offering an auditory perspective on what each layer "hears" and how it evolves through the network. Such approaches exemplify an emerging trend: there is increasing interest in understanding how these models do what they do. However, these approaches have primarily focused on probing for known concepts, an important but limited strategy. This raises a key question: what structures might we be missing by focusing only on concepts we already know to look for? To answer this, we propose an unsupervised concept *discovery* pipeline. **In summary, probing asks "do models encode X concept we already know?", while our pipeline asks "what concepts do models encode?" without supervision.**

## 2.4 Automated Concept Discovery and Evaluation

A central challenge in interpretability is to discover concepts a model has learned organically, rather than imposing them *a priori*. Recent frameworks address this across domains. Both Simon & Zou (2024) and Gujral et al. (2025) train sparse autoencoders on a protein language model (ESM-2), uncovering thousands of latent features, many aligning with known biological concepts and others appearing novel. Concept Bottleneck Models (CBMs) (Koh et al., 2020) instead constrain intermediate layers to discrete, named concepts, though typically with a hand-specified set. Recent variants invert this paradigm, using SAEs to first *discover* latent concepts and then *name* them automatically or semi-automatically (Rao et al., 2024). While developed outside of music, these principles are broader: generative music models likely encode diverse concepts without explicit labels. Automated discovery and evaluation of such features could reveal how these models "understand" music, and enable finer control over generated outputs.

## 3 Methods

Our discovery and steering pipeline is illustrated in Figure 1.

### 3.1 DATASET AND ACTIVATION EXTRACTION

The first step in our pipeline is to obtain a dataset of music clips for which we can obtain activations from the generative model. We use the *MusicSet* (Wei et al., 2024b) dataset for this, which is a collection of around 160,000 samples, most of which are $\approx$10 seconds long. MusicSet is a combination of data from MTG-Jamendo (Bogdanov et al., 2019), MusicCaps (Agostinelli et al., 2023), and MusicBench (Melechovský et al., 2023). All the source datasets are Creative Commons-licensed (though only MTG-Jamendo explicitly states that this applies to the audio). We chose this dataset because it offers a comparatively diverse range of genres, instrumentation, and musical characteristics at a larger scale than many alternatives. We also found the data to be more stylistically diverse and high-quality than FMA (Defferrard et al., 2016), another popular option. We produce activations by feeding the music into two pre-trained MusicGen (Copet et al., 2024) models: MusicGen-Large (henceforth, MGL) and MusicGen-Small (henceforth, MGS). For each, we extract activation vectors from five residual stream layers: the early layer (layer 2), late layer (second-to-last), middle layer, and layers at 25%, 50%, and 75% depth. This corresponds to $Layer_{\text{MGS}} \in \{2, 6, 12, 18, 22\}$, and $Layer_{\text{MGL}} \in \{2, 12, 24, 36, 46\}$. Activation dimensionality is 1024 for MGS and 2048 for MGL. Note: we use *unconditional* audio; see Appendix E for more details.

### 3.2 TRAINING THE SPARSE AUTOENCODERS

We train a series of SAEs to extract interpretable features from the residual stream activations of MusicGen. Each is trained to reconstruct the original activations $\mathbf{x} \in \mathbb{R}^d$ from a learned sparse latent representation $\mathbf{h} \in \mathbb{R}^{\epsilon \cdot d}$, where $\epsilon$ is an expansion factor (note: this means the latent dimension is typically *larger* than the input dimension for such SAEs).

The SAE architecture generally consists of an encoder ($\mathbf{h} = \text{ReLU}(\mathbf{W}_e\mathbf{x} + \mathbf{b}_e)$) and a decoder ($\hat{\mathbf{x}} = \mathbf{W}_d\mathbf{h} + \mathbf{b}_d$), both implemented as single linear layers where $\mathbf{W}_e \in \mathbb{R}^{\epsilon \cdot d \times d}$, $\mathbf{W}_d \in \mathbb{R}^{d \times \epsilon d}$, $\mathbf{b}_e \in \mathbb{R}^{\epsilon d}$, and $\mathbf{b}_d \in \mathbb{R}^d$ are the learned weights and biases. To enforce explicit sparsity, we apply a $k$-sparse projection operator $P_k : R^{\epsilon \cdot d} \to R^{\epsilon \cdot d}$ to $h$, yielding the sparse latent code $z = P_k(h)$. These are typically called $k$-sparse autoencoders (Makhzani & Frey, 2013; Gao et al., 2024). Where $S_k$ is the set of indices of the top-$k$ values $\in$ h:

$$z = P_k(h) = \begin{cases} h_i & \text{if } h_i \in S_k(\mathbf{h}) \\ 0 & \text{otherwise} \end{cases} \tag{2}$$

We minimize the mean squared reconstruction error, subject to the sparsity constraint. We experiment with expansion factors $\epsilon \in \{4, 32\}$ and sparsity levels $k \in \{32, 100\}$, all of which are values encountered in prior SAE literature (Gao et al., 2024; Simon & Zou, 2024).

### 3.3 FEATURE MAPPING AND FILTERING

After the SAE has trained, its latents presumably correspond to meaningful musical features. Of course, the model itself does not reveal what these are. We need some way to work backwards from these latents to objects whose content we can perceptually interpret, in relation to a particular latent dimension learned by the SAE. A conventional approach is to use *max activating examples*. In the case of language models, these are token sequences (prompts, perhaps) that most significantly activate a given feature. In the case of music, it gets a little trickier. Unlike text, tokens here represent an almost imperceptible chunk of music: about 20 milliseconds. Even with a handful of these, interpreting a feature becomes quite challenging.

To characterize the relationship between learned features and input tracks, we compute a per-track summary statistic for each feature based on its activation profile over time. Let $f_i$ denote the $i$-th learned feature, and let $\alpha_{i,j} = (\alpha_{i,j,1}, \ldots, \alpha_{i,j,T_j}) \in \mathbb{R}^{T_j \times d}$ represent its activation time series over the $T_j$ time steps of track $j$. The *mean activation* of feature $f_i$ in track $j$ is: $\mu_{i,j} = \frac{1}{T_j} \sum_{t=1}^{T_j} \alpha_{i,j,t}$. This gives us a scalar estimate of the feature's average prominence over the duration of the track.

To support downstream interpretation and analysis, we then identify a subset of features whose activation patterns suggest both selectivity and relevance. The goal is to exclude features that are either inactive, overly common, or too rare to support generalizable interpretation. Let $\delta_{i,j} \in \{0, 1\}$ denote whether feature $f_i$ exhibits nonzero average activation, which corresponds to nonzero activation at

any time step in track $j$: $\delta_{i,j} = \mathbb{I}\left(\max_{1 \le t \le T_j} \alpha_{i,j,t} > \tau\right)$ where $\tau \ge 0$ is a small activation threshold (we set $\tau = 0$). We define the corpus-level activation rate of feature $f_i$ as $r_i = \frac{1}{N} \sum_{j=1}^{N} \delta_{i,j}$ where $N$ is the number of validation tracks.

We discard feature $f_i$ if any of the following conditions hold: **(a) Inactive**: $r_i = 0$; the feature never activates in the validation set. **(b) Excessively Ubiquitous**: $r_i > \theta_{\max}$, with $\theta_{\max} = 0.25$; the feature activates in more than 25% of tracks, suggesting diffuse or polysemous behavior that limits interpretability. **(c) Excessively Obscure**: $0 < r_i < \theta_{\min}$, with $\theta_{\min} = 0.01$; the feature activates in fewer than 1% of tracks, indicating insufficient coverage to support reliable interpretation. This filtering step seeks to ensure that only features with non-trivial, non-saturated, and sufficiently frequent activation patterns are retained for subsequent analysis. Thresholds are heuristic and informed by prior work in neural feature sparsity and interpretability (Simon & Zou, 2024).

### 3.4 SELECTING EXAMPLES FOR LABELING

One challenge of labeling concepts relates to *typicality*. Extreme values (e.g., the maximum activating example) might not represent the typical set of the distribution of examples that activate a given feature. Let $E_j = \{x \mid f_j(x) > \tau\}$ denote the set of all inputs $x$ that activate feature $j$ with sufficient strength $f_j(x)$ (as before, we set $\tau = 0$ as a conservative threshold). Instead of relying on a single max-activating example, we select the top 10 highest-activating examples, $S_j \subset E_j$, and infer the feature's label from these. The choice of $|S_j| = 10$ is motivated by a balance between statistical robustness and noise from weakly activating examples. A larger activating subset improves the likelihood of capturing a representative sample from $E_j$, but diminishing returns arise as activation strength decays, especially under the sparsity constraint wherein the distribution might be truncated. Our heuristic aims to approximate the mode of the feature's natural stimulus distribution.

### 3.5 AUTOMATED INTERPRETABILITY

Now that each feature can be represented by its top-activating examples, it must be labeled. Human labeling is feasible but does not scale. With expansion factor $32\times$ on a $d{=}2048$ layer, an SAE yields $32d{=}65,536$ features; if only $\sim 1\%$ activate, that is $\approx 655$ features. At 10 examples $\times$ 10 s per feature, this is $\sim 65,500$ s ($\approx 18.2$ h) of listening for one SAE, excluding deliberation, and the cost multiplies across layers and settings. A scalable alternative is required.

Though prior automated interpretability methods use language models to explain linguistic concepts, musical concepts present potentially unevenly across much longer sequences and lack mature music-language models to use in such a setup. As such, to generate descriptive, meaningful labels for the discovered features, we use a multi-step pipeline:

1. A *generative label proposal* strategy. We query a large multimodal model (i.e., Gemini Flash 1.5, but explored others in Appendix B) with concatenated audio of the top-10 max activating examples for each feature (see prompt in Appendix H). We instruct the model to produce a set of concept tags, confidence scores for each, and an overarching label and description for the feature.

2. A *classifier-based* strategy where we extract a set of tags (see Appendix D) for each feature using pre-trained Essentia (Bogdanov et al., 2013; Alonso-Jiménez et al., 2020) audio classifier models. Multiple top predicted tags from each model are candidate labels.

3. We use CLAP (Wu et al., 2023) to compute semantic alignment between suggested labels and activating examples. This provides a quantitative measure of how well each concept label aligns with the audio content of the feature.

### 3.6 HUMAN VALIDATION

Ultimately, given the complexity of musical signals and the possible ambiguity of features, we conducted a human validation study (IRB approved) to assess labeling quality. For each feature, participants were presented (see Appendix F) with three audio examples (sampled from SAEs with high valid feature counts) and the candidate labels. They selected the label that best captured the commonality across the examples, rated their confidence in that choice, and provided an optional free-text label.

## 3.7 GENERATION STEERING

We then seek to test whether we can use discovered features to *steer* the model's generation. Steering modifies the forward pass by adding scaled feature vectors to the residual stream at the SAE's hook point. For a given SAE feature $j$, we perform steering by adding the corresponding decoder weight vector $\mathbf{W}_{d,j}$ to the residual stream activations during generation: $\mathbf{x}' = \mathbf{x} + \alpha \cdot \beta \cdot \mathbf{W}_{d,j}$ where $\mathbf{x}$ are the original residual stream activations, $\alpha \in (0, 1)$ is the steering strength, and $\beta$ is the maximum activation strength for feature $j$ computed over the feature $j$'s max. activating examples.

**Experimental Setup.** We evaluate steering using a neutral prompt "Simple melody" (as in Arad et al. (2025)) for steering strengths $\alpha = 0.0$ (unsteered baseline) and $\alpha = 1.0$ (max. steered version) for each feature to test the "steerability" of a concept. We used *MusicGen-Large* SAEs with exp. factor 32, $k \in \{32, 100\}$, and middle to late layers (24, 36, and 46), based on feature statistics.

**Feature Selection and Evaluation.** We identify the most steerable features by computing the cosine similarity of CLAP embeddings (Wu et al., 2023) for each feature using cosine similarity between the feature's top 10 activating examples and (1) the steered example, and (2) the baseline.

## 4 RESULTS

### 4.1 STATISTICS OF DISCOVERED FEATURES

We first **quantitatively** characterize the distribution of feature activation patterns across the validation set, and the impact of SAE hyperparameters on the number and nature of discovered features. More details are provided in Appendix A.

**Before Filtering** Initially, feature activations show a heavy-tailed distribution: some fire in many tracks, while most activate only rarely (see Figure 6 for details). SAE hyperparameters appear to mediate this balance, i.e. larger $k$ or expansion factor ($EF$) increase overall recovery but also changes the activation frequency distributions. This imbalance makes raw feature counts misleading and motivates our explicit filtering approach, to isolate features that are both selective and interpretable.

**After Filtering** Table 1 reports the number of features that survive filtering across models, layers, and SAE settings. For MGL, certain configurations yield thousands of retained features, while MGS rarely produces more than 100. Expansion factor also has a predictable effect: higher $EF$ values increase feature counts, reflecting the greater representational pressure imposed by the wider latent space. Interestingly, we find that this combined well with *larger* $k$, possibly suggesting that musical clips tend to have many co-occurring activating features (e.g. in different layers and at different time-steps). Early layers (e.g. L2) largely produce more features than later ones, though interpretability does not necessarily track count. The consistent difference between MGL and MGS indicates that scale does more than add parameters: it appears to alter the internal organization of representations in ways that make interpretable structure more extractable. Together, these patterns show that feature discovery depends jointly on the SAE and the underlying model's representational regime.

Table 1: Feature counts across models, layers, and SAE configurations (total 4697). For each model layer, the configuration providing the max. number of features is **bolded**.

| Exp. F | $k$ | MusicGen Large | | | | | MusicGen Small | | | | |
|---|---|---|---|---|---|---|---|---|---|---|---|
| | | L2 | L12 | L24 | L36 | L46 | L2 | L6 | L12 | L18 | L22 |
| 4 | 32 | 12 | 0 | 4 | 4 | 3 | 0 | 0 | 0 | 0 | 0 |
| 32 | 32 | 30 | 117 | 149 | **135** | 71 | 3 | 7 | **4** | 7 | 9 |
| 4 | 100 | 407 | 109 | 147 | 28 | 25 | 6 | 3 | 0 | 0 | 0 |
| 32 | 100 | **2344** | **222** | **412** | 131 | **177** | **59** | **47** | **4** | 4 | **17** |

### 4.2 EXAMPLES OF FINDING CANONICAL MUSICAL CONCEPTS

Retained features often correspond directly to established musical constructs, showing us how these are encoded within the model. Figure 2 (left) shows manually annotated examples from MGL:

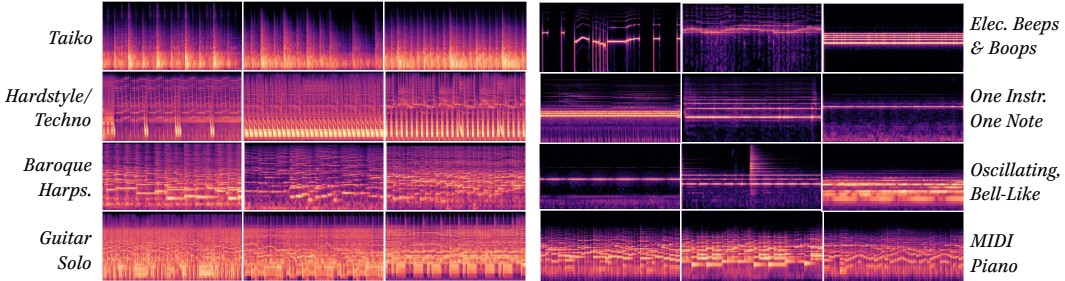

Figure 2: Examples of features discovered using the sparse autoencoders we train. Note: these examples are labeled manually. Spectrograms highlight similarities across examples within a concept.

1. **Taiko Drums** *(k=32, EF=4)*: activates on resonant Taiko and other low-pitched drums, isolating a timbral family recognizable in theory and practice.
2. **Hardstyle Techno** *(k=32, EF=32)*: consistently aligns with the hardstyle subgenre, capturing a distinct rhythmic and production signature.
3. **Baroque Harpsichord** *(k=32, EF=32)*: isolates the plucked timbre and contrapuntal texture of harpsichord repertoire, as well as similar stylistic cues in Baroque string writing.
4. **Rock Guitar Solos** *(k=32, EF=4)*: activates on electric guitar solos, integrating timbre, ornamentation, and melodic phrasing characteristic of the style.

These cases demonstrate that SAEs can discover categories that existing vocabulary already recognizes as meaningful. The match between latent features and canonical concepts indicates that the activations are not arbitrary artifacts but can and do internally encode distinctions salient to musicians and listeners in ways we can localize and recover from their internal activations.

### 4.3 EXAMPLES OF EMERGENT MUSICAL REGULARITIES

Not all features map neatly to established verbal categories. Some capture patterns that are perceptually coherent but poorly described in existing theory (Figure 2, right):

1. **Electronic Beeps and Boops** *(k=32, EF=4)*: activates on diverse synthetic tones and glitches, a class of sounds central to electronic genres but not theoretically well-defined.
2. **Single Instrument, Single Note** *(k=32, EF=32)*: isolates sustained single-note events across instruments, suggesting that the model may have detectors for basic atomic units of musical *texture*.
3. **Oscillating Bell-like Timbres** *(k=32, EF=32)*: responds to bell-*like* sounds with beating or oscillations, pointing to sensitivity to fine-grained spectral phenomena.
4. **Romantic Poppy MIDI Piano** *(k=32, EF=4)*: activates on MIDI piano in pop-ballad contexts, apparently tuned to performance artifacts like rigid quantization and compressed dynamics that co-occur with the poppy, romantic stylistic frame.

These examples surface coherent musical regularities that are not fully described by standard musical terminology likely to be found in prompts or specified in probing experiments. They show that models are not limited to reproducing canonical categories but also carve out distinctions that reflect production practices, timbral subtleties, or emergent stylistic groupings. Identifying such features provides concrete evidence that generative models encode structure beyond what human-led interpretability, such as probing, is likely to anticipate. In turn, analyzing situations in which these features arise may prove useful for building them into coherent theoretical constructs when studied in detail over time by music theorists.

### 4.4 CONCEPT REPRESENTATION

We next examine how the discovered features are distributed across layers and model scales, and how these patterns connect to prior findings.

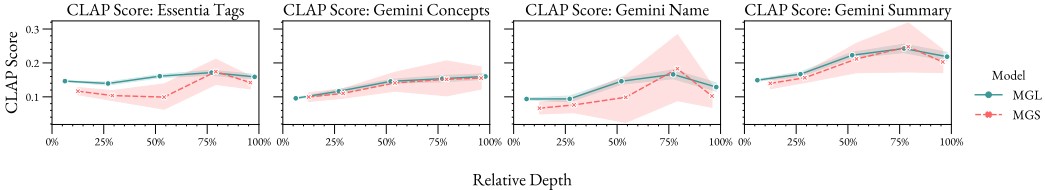

Figure 3: Avg. CLAP (Wu et al., 2023) score across layers, comparing feature audio to automatic concept labels. For MGL, later layers appear to produce more interpretable features on average.

**Do later layers encode more interpretable features?**  Figure 3 shows average CLAP scores between feature audio and their automatically generated labels, stratified by layer. For MGL, deeper layers yield higher scores, indicating that their features are more readily aligned with human-interpretable concepts. This mirrors probing results from (Ma et al., 2024), which showed that musical properties such as chords and roots are more strongly encoded in deeper transformer layers. Our results generalize that observation: later layers not only encode specific theory-driven constructs more clearly, but also produce features that can be labeled more consistently across diverse concepts.

**Does model scale change how features are organized across layers?**  To test whether larger models produce more layer-specific features, we trained a simple MLP probe to predict the layer of origin from a feature's activation profile. **Accuracy was substantially higher for MGL** ($50.29 \pm 3.41$) **than for MGS** ($40.51 \pm 3.60$)**.** This means that features in MGL are more distinctive by layer, whereas those in MGS reflect a less differentiated representational structure. These results extend probing-based findings on synthetic datasets (Wei et al., 2024a): scale not only increases the number of recoverable features, but also sharpens the division of representational roles across layers.

**How do features co-activate?**  In Appendix C, we analyze feature *co-activation* patterns, showing examples of cross-layer concept connections which emerge from SAE training.

## 4.5 Automated Interpretability

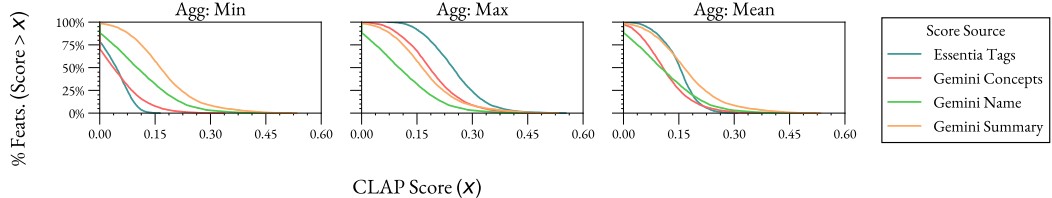

Figure 4: Distribution of max. CLAP scores across all SAEs. Pooling both Gemini- and Essentia-produced labels, we score them using CLAP, showing the trade-off between confidence and coverage at different potential filter levels.

**CLAP Scores**  We assessed label quality using CLAP (Wu et al., 2023) alignment between feature audio and assigned labels. Figure 4 shows the distribution of maximum scores across all SAEs. Essentia tags achieve stronger alignment than Gemini with considerable overlap between the two. Overall, no single labeling strategy dominates. CLAP scores also provide a practical mechanism for filtering: thresholds can be tuned to trade off label quality against feature coverage without exhaustive manual review, though the optimal cutoff may depend on corpus and task.

**Human Evaluation**  To compare methods directly, we evaluated 400 features per pipeline with 80 participants (40 per method, 10 features each). Participants chose the best label, rated confidence, and could supply alternative labels. Confidence was higher for Essentia (3.96/5; 71% >4) than for Gemini (3.19/5; 47% >4), suggesting that while multimodal LLMs can generate open-ended labels beyond fixed taxonomies, classifier-based tags may be more reliable in practice.

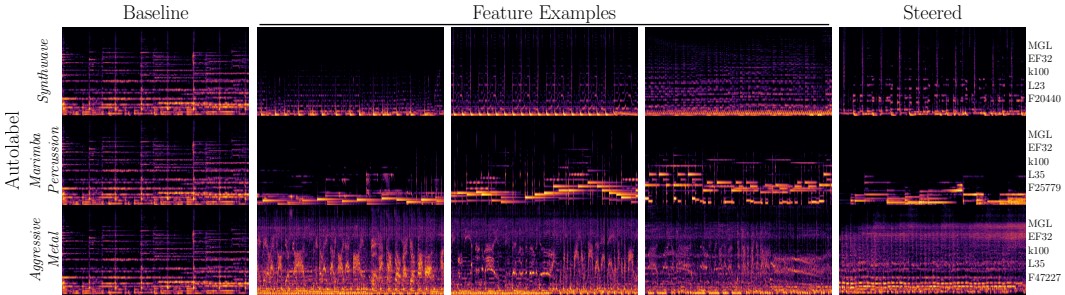

Figure 5: Examples of *steered* features. Note: these examples were labeled *automatically*. **(Baseline)** Generation without steering for "Simple melody." **(Feature Examples)** Top max. activating examples for the steering feature. **(Steered)** Generation steering with the same prompt and seed as the baseline, and maximum strength empirically calculated from the maximum activations. The steering shows close alignment with the feature examples, as seen in the spectrograms.

### 4.6 STEERING WITH DISCOVERED CONCEPTS

We next tested whether the features uncovered by our SAEs can be used for controllable generation via steering. This is a strong intervention: there is no guarantee that every discovered feature should be steerable, nor that steering will always preserve perceptual coherence. Limits to steering via SAEs have been observed in prior work (Wu et al., 2025). Thus, the goal of these experiments is to establish the *existence* of steerable concepts.

Table 4.6 summarizes results. Depending on SAE configuration, 15–35% of tested features had improved CLAP alignment with their top-activating examples under steering (with prompt "Simple Melody"). While this may appear modest, it demonstrates that a non-trivial fraction of discovered features correspond to causal, manipulable directions in activation space even under automatic labeling: actionable for controlling outputs. Figure 5 illustrates such cases, where steering shifts outputs toward the intended feature class while holding prompt and seed fixed. This establishes the potential for SAE-driven steering in controllable generation settings, though our primary goal in this work remains feature discovery (Peng et al., 2025).

Table 2: Proportion of features per SAE with positive steering improvement.

| Model | Exp. F | $k$ | Layer | Steering Improvement |
|-------|--------|-----|-------|----------------------|
| MGL | 32 | 100 | 24 | 96/408 (23.5%) |
| MGL | 32 | 100 | 36 | 46/131 (35.1%) |
| MGL | 32 | 100 | 46 | 27/177 (15.3%) |
| MGL | 32 | 32 | 24 | 44/149 (29.5%) |
| MGL | 32 | 32 | 36 | 39/135 (28.9%) |
| MGL | 32 | 32 | 46 | 16/71 (22.5%) |

We conducted a listening study with 10 participants to each listen to 10 sets of clips; each containing a baseline, a steered version, a random-direction matched-norm steered version, and a representative of the steering target, asking them to match the candidates (baseline, randomly steered, and SAE steered). Each participant's 10 sets were equally sampled from the total of top 50 steerable features. Participants largely selected the SAE-steered audio (66/100, compared to 17 each for baseline and random; $\chi^2 = 48.02$, $p < .0001$). This suggests that the steering effects are clearly perceptible.

## 5 CONCLUSION

Our results demonstrate the feasibility and potential of using sparse autoencoders to discover interpretable concepts in large music generative models, i.e. using models for *musical discovery* (Singh et al., 2024). We have shown that these models learn a rich set of musical patterns, ranging from well-established concepts to emergent regularities that warrant further investigation. Through future work in this area, we believe it is possible to build a comprehensive understanding of how neural networks represent and generate music. Our pipeline offers a scalable path towards this goal.

ACKNOWLEDGMENTS

We gratefully acknowledge AWS for providing computational resources through the AWS Research and Engineering Studio (RES) pilot program. We thank Brian McCarthy and Jared Novotny for their valuable support, and Peter Fisher (MIT ORCD) for making this collaboration happen. The authors acknowledge the MIT Office of Research Computing and Data for providing high performance computing resources that have contributed to the research results reported within this paper. MC is supported by a fellowship from "la Caixa" Foundation (ID 100010434) with code LCF/BQ/EU23/12010079. We also thank Michael Casey, Anna Huang, and the HAI-Res lab for supportive comments.

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

## A    ADDITIONAL RESULTS: STATISTICS OF DISCOVERED FEATURES

Figure 6 presents feature activation statistics. For each SAE, we plot the relationship between the average activation strength of each feature (y-axis) and its prevalence, as defined by the fraction of the validation set for which the feature's mean activation is positive (x-axis). Figure 7 visualizes the results from Table 1 for an alternate view, to facilitate overall comparisons.

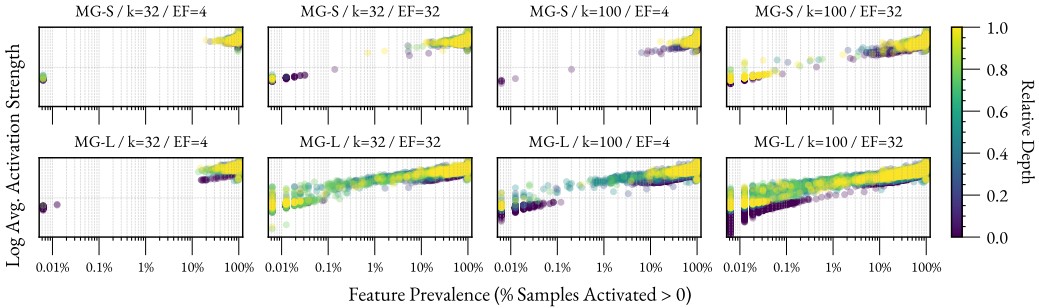

Figure 6: Distribution of learned feature activation characteristics across the MusicSet validation cohort, prior to filtering, for various SAE configurations (log-log plot). The x-axis shows feature prevalence (fraction of validation tracks on which a feature exhibits non-zero mean activation). The y-axis indicates mean activation strength. The observed heavy-tailed distributions, with many features exhibiting either very broad or very sparse activation patterns, suggest the need for a more principled filtering method.

## B    CHOICE OF CAPTIONING MODEL

We compared several versions of Gemini models as a supplementary analysis, specifically *Flash 1.5*, *Flash 2.0*, *Flash 2.5*, and *Pro 2.5*, as automated labeling pipelines. We conducted a preliminary evaluation of label quality using the CLAP score (Wu et al., 2023).

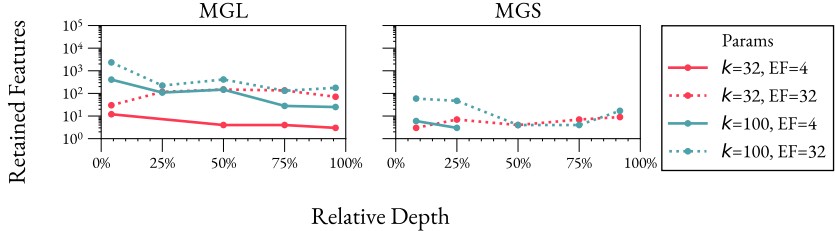

Figure 7: Impact of SAE hyperparameters and target model on yield of viable musical features after filtering. Each bar represents the number of features retained for a specific configuration, varying by MusicGen model size (small or MGS/large or MGL), target layer, $k$-sparsity, and expansion factor ($EF$). The substantial variation in feature counts shows how these parameters affect discovery of statistically robust features.

Figure 8 shows average CLAP scores per MusicGen layer across models. Figure 9 presents the ECDF of maximum CLAP scores over all labeled features. Surprisingly, Gemini Flash 1.5 consistently produces the highest CLAP alignment on the whole, despite being the oldest model tested.

Given these results, although we acknowledge there are significant limitations arising from using CLAP as the evaluation metric, we use Flash 1.5 as the default captioning model for all automatic labels in the paper's primary analyses.

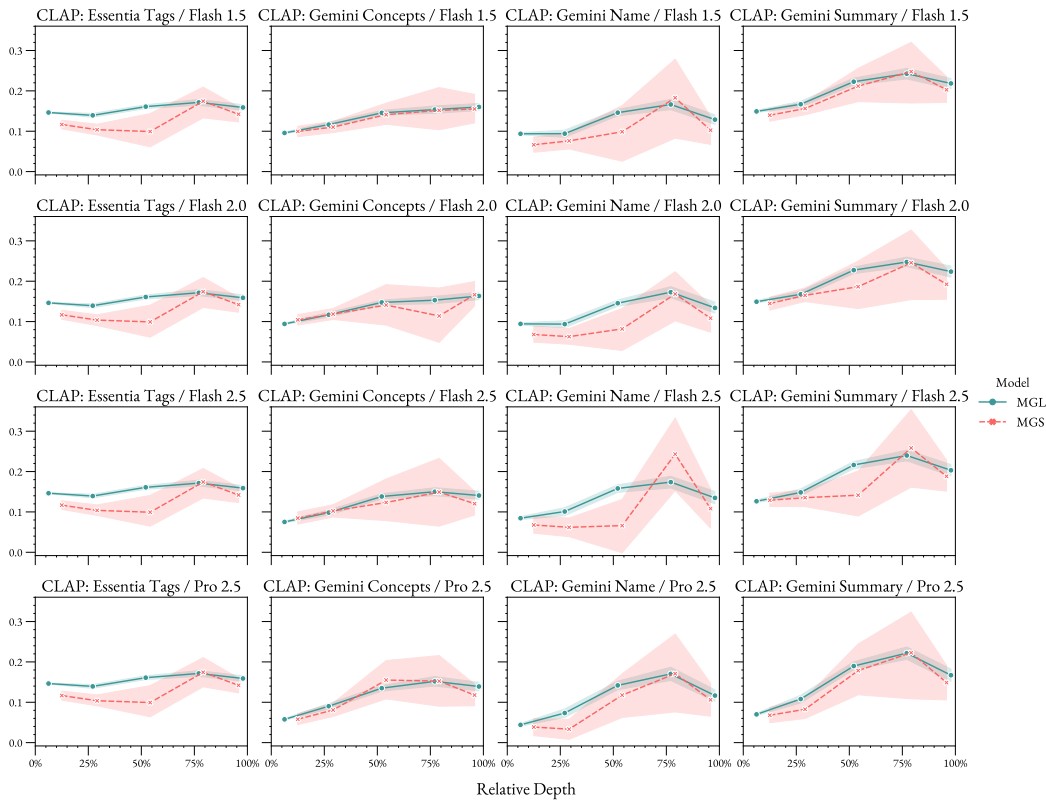

Figure 8: Average CLAP scores by MusicGen layer for each Gemini model used as the captioning pipeline. Gemini Flash 1.5 achieves the highest alignment on average.

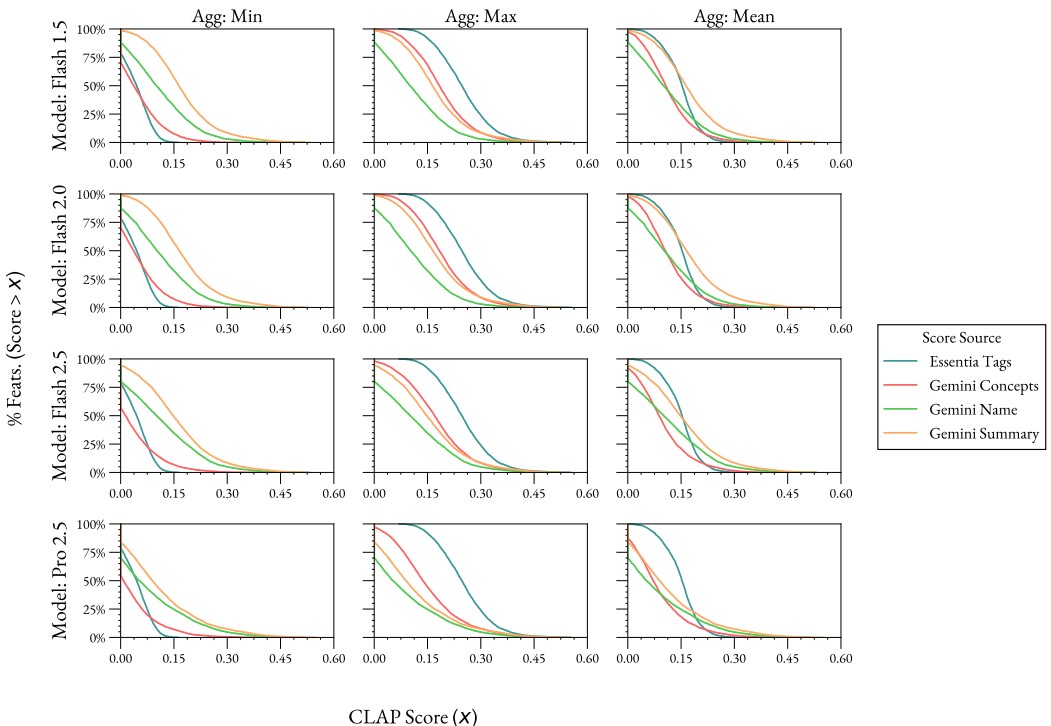

Figure 9: ECDF of maximum CLAP scores across all features for each Gemini variant. Flash 1.5 yields a higher proportion of strong alignments than newer models.

## C  FEATURE CO-ACTIVATIONS

We conducted large-scale measurements of *feature co-activation* across models, layers, and SAE configurations in order to understand when distinct learned features might activate on the same musical input. For a pair of features $(f_i, f_j)$, we treat the number of tracks common among their respective top-10 activating examples as an indicator of functional relatedness. Although the most common non-zero outcome is that two features overlap on exactly one track, the long tail of rarer, higher-overlap cases provides (observational) evidence of meaningful structure. Here we summarize the most clear and recurring qualitative patterns observed in this tail.

**Same-site convergence across distinct SAEs.**   First, we find cases where separate SAEs trained on the same model recover nearly identical concepts at the same layer position. Representative examples include:

- **Distorted Rock** (*MGL L36*) by three separate SAE configurations: $k = 32, EF = 32; k = 100, EF = 32; k = 100, EF = 4$
- **String Orchestra** (*MGL L2*) by $k = 100, EF = 4; k = 100, EF = 32$
- **Reverberant Vocals and Percussion** (*MGS L22*) by $k = 100, EF = 32; k = 32, EF = 32$

These patterns indicate that certain musical concepts may be encoded robustly enough in the model's residual stream that different SAEs can reliably recover them.

**Cross-layer hierarchical relationships.**   We also observe cases where early-layer features and deeper-layer features appear to express related musical ideas at different (e.g. increasing) levels of abstraction. For example:

- **Wind-Dominated Folk Drone** (*L12*) → **Eastern European Folk Wind Music** (*L36*) in *MGL* $k = 32, EF = 32$ with 50% overlap between features (5/10 examples)

- **Jazzy Brass** (*L24*) → **Brass and Syncopation** (*L45*) in *MGL* $k = 100, EF = 4$ with 50% overlap

- **Arpeggiated Electronic Music** (*L6*) → **Synth Pop** (*L18*) in MGS $k = 32, EF = 32$ with 40% overlap

Such patterns point to potentially semantically hierarchical pathways in the learned features, where deeper layers might develop earlier features.

**Overall structure.** Across the full set of experiments, all configurations of MGL and MGS exhibit the same broad pattern: most feature pairs co-activate sparsely if at all (see right side of Figure 10). Figure 11 shows statistics of features co-activating *across layers*. These results suggest that certain SAE configurations are more successful at recovering these than others. For the most successful one, we show layer co-activation statistics in the heatmap to the right (within-layer co-activations are zeroed out here). Co-activations are distributed rather than e.g. locally biased. Finally, in Figure 12, we examine to what degree different SAE configurations can recover similar features in similar locations. Here, there is a split between cooccurrence rates and strengths: *MGL L2* has by far the most cooccurring features across different SAE configurations, but the lowest average strength. In contrast, *MGS L22* has among the fewest cooccurring features, but the highest average strength. This could be due to, for instance, the degree of compression within the model and layer: *L22* is the latest layer we test in the small-scale *MGS* and may have more well-formed features that are consistently recovered, while *L2* is the earliest layer we test in the large-scale *MGL*, and could have more generic and low-level features. We note however that these are observations rather than formal, mechanistic findings.

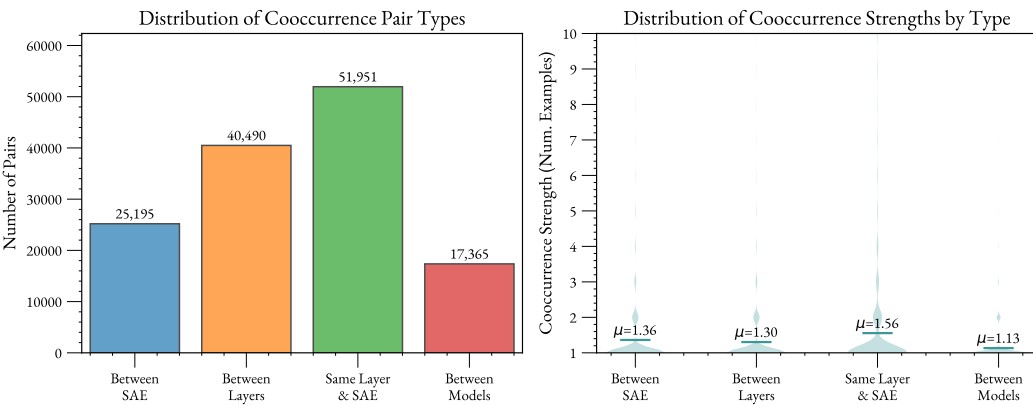

Figure 10: Co-activation/cooccurrence statistics by unit of analysis: between SAE configurations on the same model/layer, between layers on the same model and SAE configuration, within the same layer/model/SAE configuration, and between MGS and MGL models.

In aggregate, these observations suggest that while the bulk of discovered features behave independently, there is a nontrivial amount of structure learned across different parts of the models that co-activation analysis can help to reveal. A deeper analysis might look at how these observations change under different co-activation strength thresholds, however we acknowledge this requires a much larger scope of experimentation.

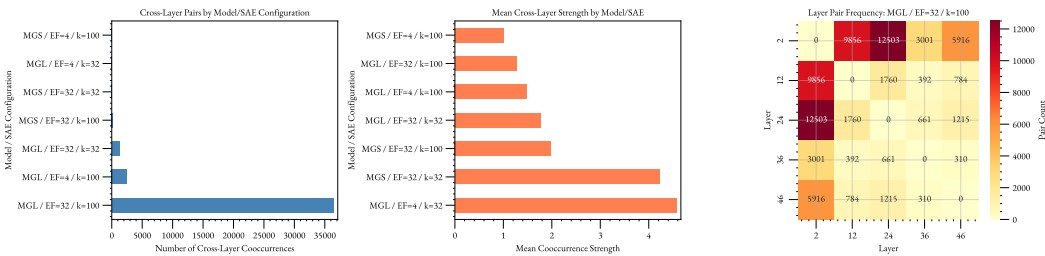

Figure 11: Statistics of cross-layer co-activations.

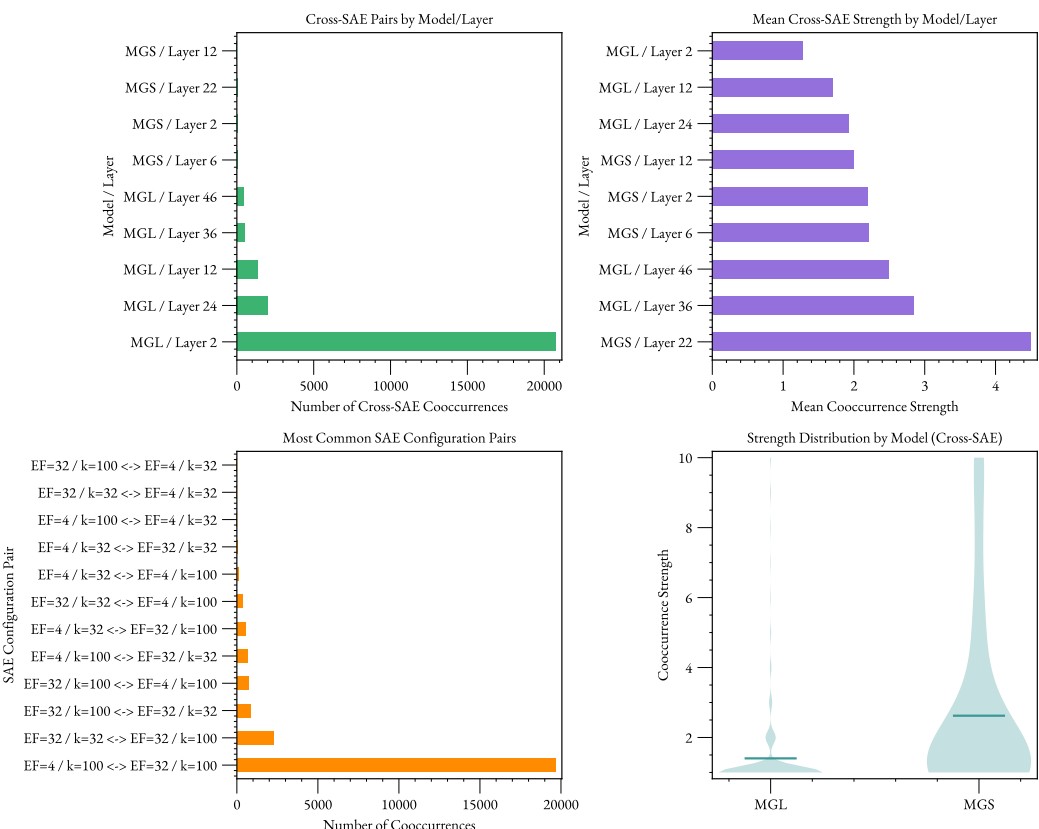

Figure 12: Statistics of co-activations across SAE configurations, within the same model and layer.

## D ESSENTIA TAGS

### ESSENTIA MODEL TAGS

We use the following Essentia feature extractors and tagging models. By default, we select the top 3 predicted tags from each model, since we observe too much noise in the results to rely solely on the top predicted tag and these are often used in multi-label settings (exceptions are noted and justified below):

- **Genre-Discogs400**: Genre prediction (top 3)
- **Mood-Jamendo**: Multi-label mood/theme (top 3)

- **Instrument-Jamendo**: Instrument presence (top 4, to account for common band compositions)
- **Tags-Jamendo**: General music tags (top 3)
- **Tags-MSD**: General music tags, different set (top 3)
- **Moods-Audioset**: Mood categories, different set (top 2; these tend to be less distinct)

These in turn depend on VGGish, EffNet, or MusicCNN embeddings, all of which are also extracted.

## E  PROMPT-CONDITIONED ACTIVATIONS

Our primary focus in this work was to analyze the musical representations that emerge *without* text conditioning. These representations are especially important because they capture how the model organizes musical structure in its own representational space, rather than simply responding to verbal instructions. Whereas prompts are already verbal and thus relatively easy to align with human categories, internal musical features may be tacit, difficult to describe, and poorly characterized in existing theory. By targeting this layer of representation, we aim to surface structures that extend beyond what can be easily named, and therefore hold greater potential for discovery.

We see text-conditioned analysis as a valuable future direction, but our findings suggest that it is not tractable under current conditions. First, open datasets with paired text–music examples remain limited. Captions are often sparse, noisy, or stylistically inconsistent, and prompt–audio alignment is insufficient to support reliable, large-scale analysis. This means that conclusions about text-conditioned representations would be confounded by dataset artifacts rather than model behavior. Second, from a methodological standpoint, isolating the contribution of text requires ablation and control techniques that go beyond existing ones. Mechanistic interpretability has not yet converged on reliable tools for partitioning or disentangling multi-modal conditioning, making this a substantial open problem.

Addressing these challenges would require both new datasets with carefully designed prompt–audio correspondence and new methodological advances to interpret conditioning effects in multimodal models. We believe this is a promising research agenda, but it demands significant additional work beyond the scope of this paper. Our contribution here is therefore to establish a foundation by studying the non-textual musical representations directly, which offers both theoretical value, in revealing the categories that emerge independently of language, and practical value by providing a baseline for future multimodal interpretability.

## F  INSTRUCTIONS GIVEN TO PARTICIPANTS

> **Study Details**
>
> We are assessing the quality of a novel method for discovering interpretable concepts in generative music models. You will be presented with a series of short sounds with a common pattern or feature. Given a set of options, you will be asked to select the one that best describes all the sounds, as well as your confidence in your prediction.
>
> [FOR EACH SAMPLE]
>
> Select the closest category representing what all these sounds have in common:
>
> [Audio 1] [Audio 2] [Audio 3]
>
> [Option 1]
> [Option 2]
> [Option 3]
> [Option 4]
> [Option 5]
>
> How confident are you that this label describes the common feature(s) between these well?

[Completely confident]
[Fairly confident]
[Somewhat confident]
[Slightly confident]
[Not confident at all]

Write down your own label for what these sounds have in common musically

[FREE TEXT]

## G   CONNECTIONS TO MUSIC THEORY

This work is conceptually related to several core threads we are aware of in music theory, cognitive musicology, and computational modeling of music structure. Here, we clarify how our approach relates to existing work by seeking to uncover "operational" structures (what we refer to as features or concepts) in generative models.

**Statistical foundations of music theory**   Temperley (2004; 2007) argues that listeners develop an understanding of many core elements of Western music (such as meter, harmony, and voice leading) by internalizing probabilistic regularities encountered in musical corpora. From this perspective, foundational concepts within music theory, rather than being seen solely as elements of a formal axiomatic system, can be understood as principles that often reflect this inductively learned, tacit knowledge of musical likelihoods. Temperley's computational cognitive models are themselves compact and idealized representations which help explain why these music-theoretic constructs are perceptually and musically relevant.

Our approach parallels this in an AI setting. Rather than analyzing human musical corpora directly to infer cognitive principles, we seek to "reverse-engineer" generative models *trained* on such data to recover the internal distinctions and abstractions they have formed. The concepts discovered in our pipeline reflect these model abstractions, many of which correspond to statistical regularities observed in music (e.g. genre, texture, instrument timbre, harmonic style).

**Operational concepts vs. formal theory**   Gjerdingen's schema theory (Gjerdingen, 1990; 2007) discusses the role of learned patterns in shaping musical fluency. His analysis of Galant phrase structure proposes that composers relied on a rich set of prototypical configurations (or *stock musical phrases* in particular), acquired through exposure and practice rather than via formal rule systems. These schemata were typically only formalized retroactively through pedagogical and analytical discourse, as we imagine concepts of the type our pipeline discovers might be in the future. As such, our work focuses on *proto-theoretical regularities*: we extract features that appear to be functionally meaningful to the model regardless of whether they map cleanly to any established theoretical category. Some features do align with known "schemata" but others do not, yet still display internal coherence. In this sense, our pipeline seeks to unearth raw material from which a theory (either of a model's music understanding, or of some aspect of music itself) could potentially be formed.

**Meta-theoretical implications**   This empirical approach may also serve as a testing ground for meta-theoretical questions about music theory. If large-scale generative models learn internal representations that resemble certain theoretical concepts but diverge from others, this may point to systematic gaps or biases in how theory abstracts from musical practice. Conversely, the emergence of coherent but non-canonical features suggests that models may encode distinctions that are musically significant but neglected in prior analyses. In this way, our work does not propose a theory of music, but can be seen as contributing a method for investigating the gap between musical practice and theoretical formalism. We propose that such discovered features features be treated as *hypotheses* about musical structure that may be refined, validated, or rejected through future research.

# H  GEMINI DETAILS

Our prompt to automatically label concepts using Gemini's API, developed through iterative testing:

---

**Prompt for Gemini's Labeling**

*Listen very carefully to this set of audio clips, which consists of song snippets concatenated in random order.*

*You need to discover common musical patterns across the whole set, to identify what musical feature is shared across all clips. You will need to listen carefully. For each potential concept you identify, output a name, a confidence score between 0 and 1 (where 1 is highest confidence), and a concise description of the concept.*

*At a higher level, describe the overall concept shared across the set, give it a suitable name, and provide an overall confidence score (0 to 1).*

*Describe the \*underlying concepts\* not the specific audio snippets (e.g. your description could say "the concept" but not "the audio snippets"). However, try to avoid such verbiage altogether and concisely describe the musical concept's main attributes.*

*Include NO FILLER text.*

*Focus on being specific. Concepts could relate to genre (e.g., hip-hop, salsa, reggaeton, balkan), instruments (e.g., piano, cello, guitar, flute), recording/production techniques (e.g., reverberation, drones, noise, DJ scratching, beatboxing, drum machine, hi-hat patterns, fingerpicking, live recording artifacts, low-pass filtering), or more nuanced musical ideas (e.g., drum solo, chill dance rhythm, serene woodwinds arrangement). These are illustrative examples, NOT a fixed list to choose from.*

---

We use the following model definition to constrain Gemini's response via structured output generation:

---

**Gemini's Response Schema**

```python
from pydantic import BaseModel, Field

class Concept(BaseModel):
    """Represents a single identified musical concept."""
    name: str = Field(..., description="Concise name for the musical
        ↪ concept.")
    confidence: float = Field(..., ge=0.0, le=1.0, description="
        ↪ Confidence score (0.0 to 1.0).")
    description: str = Field(..., description="Brief description of
        ↪ the concept.")

class ConceptLabels(BaseModel):
    """Represents the overall analysis result for a set of audio clips
        ↪ ."""
    concepts: list[Concept] = Field(..., description="List of specific
        ↪  concepts identified.")
    overall_summary: str = Field(..., description="Overall description
        ↪  of the shared musical concept [no full sentences, concise
        ↪ summary of underlying concept, ignore snippets].")
    overall_name: str = Field(..., description="Concise name for the
        ↪ overall shared concept.")
    overall_confidence: float = Field(..., ge=0.0, le=1.0, description
        ↪ ="Overall confidence score (0.0 to 1.0).")
```

---

## I    COMPUTE RESOURCES

All model training runs in this paper run on a node with 4x NVIDIA L40s GPUs, as well as some of the experiments (e.g. CLAP score computation). Other experiments use a CPU compute node with 128 Intel Xeon Platinum 8375C CPUs @ 2.90GHz each for parallelization.

## J    LLM USE DISCLOSURE

We used large language models for minor copy editing, including improving grammar and phrasing. The authors reviewed all changes.

