# OpenReview forum: "Discovering and Steering Interpretable Concepts in Large Generative Music Models"
_ICLR.cc/2026/Conference — ICLR 2026 Poster_

### Official Review · Reviewer_PtMu · 2025-10-28

**Soundness:** 3
**Presentation:** 4
**Contribution:** 4
**Rating:** 8
**Confidence:** 4

**Summary:**

This paper presents a comprehensive framework for uncovering and manipulating interpretable latent features in pretrained music generation models.
The authors apply sparse autoencoders (SAEs) to the residual streams of MusicGen, discovering sparse “concept vectors” that correspond to meaningful musical dimensions (e.g., timbre, rhythm, genre).
The framework further proposes automatic labeling of each discovered feature via LLM-based captioning and MIR taggers, and demonstrates preliminary steering by injecting these features back into the model.

Overall, the work is technically sound, conceptually elegant, and well-motivated, providing a significant step toward mechanistic interpretability in multimodal (music–audio–text) systems.

**Strengths:**

Novel application of mechanistic interpretability to music.
The use of SAEs to analyze music model activations is both original and conceptually clean. It connects recent interpretability work (e.g., Anthropic’s neuron-level SAE studies) to a new and highly complex modality.

Bridging analysis and control.
The proposal to “steer” MusicGen by manipulating discovered features introduces an exciting bridge between interpretability and controllability—potentially enabling a new paradigm of interpretable generation.

Strong conceptual framing.
The paper explicitly situates itself within the context of “unsupervised concept discovery,” distinguishing itself from traditional probing methods that rely on predefined human labels. This distinction is theoretically well-grounded and clearly communicated.

**Weaknesses:**

Steering demonstration missing.
The paper’s most exciting claim—that discovered features can steer MusicGen—remains theoretical.
A simple demo (e.g., modulating one feature and showing changes in audio or spectrogram) would transform this from a conceptual claim to a verifiable capability.

Emergent ability needs stronger evidence.
To argue that new control dimensions have emerged, the authors should show that these SAE-derived features are not trivially mappable to existing models such as CLAP or other audio–text embeddings.
In other words, the learned features should correspond to new, controllable factors that cannot already be directly used to condition MusicGen or described by current models.
This would validate that the SAE uncovers genuinely new interpretable concepts rather than re-labeling known ones.

(my understanding is that because the automatic labeling step relies on existing music/audio to text models, the performance is somehow bounded by their capability. E.g., if CLAP cannot do a good job on recognizing the emotion of music, the proposed model won't have good emotion interpretability. That said, novel combinations of labels are very possible, and such combinations cannot be directly used as the text prompt of musicgen. )

happy to adjust the rating if these issues are resolved.

**Questions:**

1 Could you include an explicit steering demo—for instance, modifying one SAE latent and showing corresponding audio differences or spectrogram shifts?
2 could you show "strong emergence" in the sense that the learned concept cannot be directly used as text prompt to steer MusicGen (show that doing this will lead to poor results); also CLAP cannot output such a concept.

---

> ### Author Response · Authors · 2025-11-22
>
> We’re really grateful for your feedback and the opportunity to improve this paper. Thank you for recognizing the novelty, steerability, and strong conceptual framing.
>
> > Steering demonstration missing. The paper’s most exciting claim—that discovered features can steer MusicGen—remains theoretical. A simple demo (e.g., modulating one feature and showing changes in audio or spectrogram) would transform this from a conceptual claim to a verifiable capability.
>
> Thank you for this suggestion. We have constructed such a demo with spectrograms in Figure 5, but welcome any suggestions for additional presentation strategies. We will add the corresponding steering audio to our public feature dashboard.
>
> Following 1vEC’s suggestion, we also conducted a listening study via Prolific where we recruited 10 participants to each listen to 10 sets of audio clips, where each set contained a baseline, a steered version, a random matched-norm steered version, and a representative of the steering target, asking participants to match the candidates (baseline, randomly steered, and SAE steered). Each participant’s 10 sets were equally sampled from the total of top 50 steerable features. Participants largely selected the SAE-steered audio (66/100, compared to 17 each for baseline and random; $\chi^2 = 48.02,\ p < .0001$). This suggests that the steering effects are clearly perceptible.
>
> > Emergent ability needs stronger evidence. To argue that new control dimensions have emerged, the authors should show that these SAE-derived features are not trivially mappable to existing models such as CLAP or other audio–text embeddings.
>
> Here, we note that it’s not necessary that such control dimensions are not mappable to CLAP or other embeddings; the difficulty is *finding* them. For example, let us take the example of “One Instrument, One Note.” To find this in CLAP, one would need to (1) think of this concept, (2) get the text embedding from CLAP, (3) search over the activation space of MusicGen to discover which perturbations correlate with increasing CLAP score with this prompt. This would be extraordinarily difficult, and still fundamentally relies on supervision (knowing which concept to search for).
>
> In other words, CLAP can *evaluate* this label and associated examples once they are discovered, but it cannot *discover* them, which is the main distinction: our pipeline produces perceptually coherent directions in the activation space without requiring supervision.
>
> > my understanding is that because the automatic labeling step relies on existing music/audio to text models, the performance is somehow bounded by their capability.
>
> We agree! This is a limiting factor for the strength of the pipeline, but we argue this will improve as audio/music captioning models continue to improve, since captioning is a task the audio/music community aims to make progress on [e.g. 1-3].
>
> [1] Liu, Shansong, et al. "Music understanding llama: Advancing text-to-music generation with question answering and captioning." ICASSP 2024-2024 IEEE International Conference on Acoustics, Speech and Signal Processing (ICASSP). IEEE, 2024.
>
> [2] Deng, Zihao, et al. "Musilingo: Bridging music and text with pre-trained language models for music captioning and query response." Findings of the Association for Computational Linguistics: NAACL 2024. 2024.
>
> [3] Lanzendörfer, Luca A., et al. "Bootstrapping Language-Audio Pre-training for Music Captioning." ICASSP 2025-2025 IEEE International Conference on Acoustics, Speech and Signal Processing (ICASSP). IEEE, 2025.
>
> > Could you include an explicit steering demo
>
> We have added a set of steering examples at the top of the anonymous webpage linked in the paper, with audio and spectrograms.
>
> > could you show "strong emergence" in the sense that the learned concept cannot be directly used as text prompt to steer MusicGen (show that doing this will lead to poor results); also CLAP cannot output such a concept.
>
> On the same dashboard, we also include one such case study: a feature for quiet/silence which, despite extensive prompt engineering effort, we were not able to get MusicGen to output. This may be related to a recently observed phenomenon that concepts models are able to produce are not necessarily able to be found through human steering [4]. Re: CLAP, as we previously noted, CLAP can only evaluate text-audio alignment; it has no power to output any concepts.
>
> [4] Vafa, Keyon, et al. "What's Producible May Not Be Reachable: Measuring the Steerability of Generative Models." arXiv preprint arXiv:2503.17482 (2025).

---

> > ### Author Response · Authors · 2025-11-27
> >
> > Thank you again for your reviewing effort, and for appreciating our work. We would love to clarify anything that might help with your evaluation before the deadline early next week. If all questions have been addressed, we would appreciate it if you could raise the score as you had mentioned.

---

> > ### Comment · Reviewer_PtMu · 2025-11-27
> >
> > I listened to the new steering demos and they convincingly address my earlier concern.
> > The key insight—finding meaningful latent directions—is indeed important, and this work is clearly more interpretable than CLAP-style embeddings.
> >
> > That said, there is a subtle distinction between interpretability and controllability. Turning an interpretable feature into an actionable control still requires post-hoc labeling or supervision, and the ”extraordinarily difficult” tedious pipeline the authors describe is essentially how CLIP- or CLAP-based systems implement control today when coupled with an LLM.
> > .
> > If both systems aim to steer along the same conceptual dimensions, an interesting open question is whether the proposed SAE representation would actually yield better or more precise control than those supervised, prompt-driven methods.

---

> > > ### Author Response · Authors · 2025-11-28
> > >
> > > Thank you for your quick response and continued engagement; we really appreciate it. Thanks also for listening to the steering examples.
> > >
> > > Although the discussion has concluded, we wanted to be sure to respond to your last comment.
> > >
> > > We agree re: the important distinction between interpretability and controllability. This paper is very much focused on the former (interpretability), with the controllability as supplementary evidence for the potential of concept discovery. We believe it is important future work to improve the steerability to compare with supervised methods (and address the open question which you mention), but is out of scope of the core enabling contribution we aim for here.

---

### Official Review · Reviewer_eAhf · 2025-10-31

**Soundness:** 2
**Presentation:** 3
**Contribution:** 2
**Rating:** 4
**Confidence:** 3

**Summary:**

I am a little confused by the premise of this paper, and would appreciate it being explained in the rebuttal (in which case I will be most willing to revise my score).  It seems that the paper is arguing that there are concepts humans don’t have words for and that “Electronic Beeps and Boops” is such an example.  Yet the authors are validating using both a multimodal audio -> text model (gemini) and CLAP, which seems to presuppose (and, as I have seen it used, correctly) that CLAP itself can map “Electronic Beeps and Boops” and audio to a valid score.  If one defines theory as “the things that show up in 20th century textbooks”, “beeps and boops” wouldn’t be there (for more sociological/historical than cognitive reasons), but I wouldn’t label them as “concepts we don’t have names for”, particularly given the success with CLAP.

But to summarize the paper's technical contributions - they have used sparse autoencoders (SAE's) to infer common patterns of activation.  They use this in the context of interpretability by using Gemini to provide labels for these common patterns, do some basic analysis of these data patterns, and try to steer the MusicGen model using these patterns (although it's unclear if that was successful or not).

**Strengths:**

This paper engages with the interpretability literature to provide a lens into the relationship between a highly complicated model and the sort of language we use as humans to describe music.  This provides much needed research at the intersection of model interpretability and music generation.  They successfully determine that they can use SAE's to uncover important information about how music data is structured, including not only the fact that there are such labels, but that they are in some instances highly correlated with each other.

**Weaknesses:**

1. I am unconvinced by the contributions of this paper.  Fundamentally, the claim that they are producing concepts we "don't have a name for" seems blatantly false given that they use a multimodal model to label them (unless I'm misunderstanding something).

2. The numbers for the steering example don't seem impressive without more information - having "any improvement at all" on 20-25% of features compared to None could be noise.

3. For a paper about music, it is very limiting to take the "mean activation" of a feature over 10 seconds.  For instance, unless I'm misunderstanding, the example they gave about chord progressions being something we do have interpretable concepts for, \textit{unlike} "beeps and boops", actually reveals a significant flaw in their approach - I don't think it would be mathematically possible to infer a chord progression feature from mean activations over 10 seconds.

**Questions:**

1. Do you have any way of measuring “degree to which this is a previously untheorized label”?

2. On line 288 – could you elaborate on what the “baseline” is?  Uncontrolled musicgen?

3. On line 310 – could you possibly provide examples of co-occurring activating features, and mention if there were any there that were unexpected?  I’d imagine that determining cooccurrences might be more enlightening than the concept names themselves.

4. Can you clarify the diagram on line 385?

5. Can you speak as to how your features are different than the features CLAP can recognize, or why your contribution is an improvement over just having access to CLAP?

6.   Could you describe the essentia tags used?

---

> ### Author Response · Authors · 2025-11-22
> **[1/2]**
>
> We’re really grateful for your effort in reviewing our work. Thank you for recognizing that this is a novel and timely problem in interpretability.
>
> > Fundamentally, the claim that they are producing concepts we "don't have a name for" seems blatantly false given that they use a multimodal model to label them (unless I'm misunderstanding something).
>
> We believe this is a misunderstanding. We use a multimodal model to label features, but *it does not succeed at labeling such features*. The labels we suggest for them are manually specified (as noted in the paper).
>
> > The numbers for the steering example don't seem impressive without more information - having "any improvement at all" on 20-25% of features compared to None could be noise.
>
> We also conducted a listening study via Prolific where we recruited 10 participants to each listen to 10 sets of audio clips, where each set contained a baseline, a steered version, a random matched-norm steered version, and a representative of the steering target, asking participants to match the candidates (baseline, randomly steered, and SAE steered). Each participant’s 10 sets were equally sampled from the total of top 50 steerable features. Participants largely selected the SAE-steered audio (66/100, compared to 17 each for baseline and random; $\chi^2 = 48.02,\ p < .0001$). This suggests that the steering effects are clearly perceptible.
>
> > For a paper about music, it is very limiting to take the "mean activation" of a feature over 10 seconds.
>
> This is indeed a limitation. We attempted a duration-based approach instead (average active duration, rather than average activation strength), but found these to be highly correlated. While it is possible to construct many other measures (e.g. periodicity, moments, etc.) these are difficult to generalize across different features and samples (e.g. tempo changes periodicity), which makes using other such measures much less principled.
>
> We maintain that the mean activation is thus the most sensible baseline approach to implement here.
>
> > Do you have any way of measuring “degree to which this is a previously untheorized label”?
>
> We do not, and we acknowledge that this is fundamentally difficult to precisely measure. This is a necessarily qualitative result, since the labels for such features are not generally producible automatically. Note that we do not claim the labels themselves are untheorized, but rather the features do not map to well-established theoretical terminology:
> “analyzing situations in which these features arise may prove useful for building them into coherent theoretical constructs when studied in detail over time by music theorists”
>
> For instance, it is difficult to query any existing model for “the quality that unifies [a given collection of] diverse electronic textures” and it would normally be a human act of interpretation to conceptualize such a quality, but our pipeline surfaces features that correlate with such qualities perceptually.
>
> > On line 288 – could you elaborate on what the “baseline” is? Uncontrolled musicgen?
>
> Here, the baseline is the *unsteered* prompt-conditioned output as mentioned in the paper:
>
> - “For each feature, we generate an unsteered baseline ($\alpha = 0$)”
>
> - “**(Baseline)** Generation without steering for ``Simple melody.’’”
>
> We have also added a new baseline with random direction as mentioned above.

---

> > ### Author Response · Authors · 2025-11-22
> > **[2/2]**
> >
> > > On line 310 – could you possibly provide examples of co-occurring activating features, and mention if there were any there that were unexpected? I’d imagine that determining cooccurrences might be more enlightening than the concept names themselves.
> >
> > Thank you for this insightful suggestion. We have conducted a large-scale analysis of cooccurrences **detailed in Appendix C**, which produced several interesting results. We examine co-activations using max activating examples along various dimensions: within an SAE/model pair at e.g. different layers, across SAEs on the same model, and other configurations.
> >
> > Overall, most co-activations are weak (1 example for which both activate significantly). However, looking at the long tail of strong co-activations, some interesting patterns can be observed. We offer a quick summary of the qualitative findings below:
> >
> > First, we find evidence that different SAEs recover the same feature in the same location. For example:
> >
> > * “Distorted Rock” is recovered at MGL layer 36 by three separate SAE configurations: $EF=32, k=32; k=100, EF=32; k=100, EF=4$
> >
> > * “String Orchestra” at MGL layer 2 by $k=100, EF=4; k=100, EF=32$
> >
> > * “Reverberant Vocals and Percussion” at MGS layer 22 by $k=32, EF=32; k=100, EF=32$
> >
> > We also find evidence that SAEs learn related concepts across layers, in a way that may be semantically hierarchical:
> >
> > * “Wind-Dominated Folk Drone” (Layer 12) $\to$ “Eastern European Folk Wind Music” (Layer 36), MGL $k=32, EF=32$, 50% overlap between features (5/10 examples)
> >
> > *  “Jazzy Brass” (Layer 24) $\to$ “Brass and Syncopation” (Layer 45), MGL $k=100, EF=4$, 50% overlap
> >
> > * “Arpeggiated Electronic Music” (Layer 6) $\to$ “Synth Pop” (Layer 18), MGS $k=32, EF=32$, 40% overlap
> >
> > Note: we use the automatic labels for the notes here.
> >
> > > Can you clarify the diagram on line 385?
> >
> > The figure in question (Fig. 3) summarizes the CLAP alignment between each feature's audio and its automatically generated labels. The purpose here is to illustrate an empirical trend, namely that deeper residual-stream layers tend to yield features whose associated audio aligns more consistently with their labels than the earlier layers.
> >
> > > Can you speak as to how your features are different than the features CLAP can recognize, or why your contribution is an improvement over just having access to CLAP?
> >
> > CLAP fundamentally cannot recognize features. Given both an audio clip and a textual label, CLAP can provide an estimated semantic similarity between the two. The problem solved by SAEs is an epistemic one, i.e. unsupervised identification of a coherent internal structure. Our pipeline then follows this up with automatic interpretability. Thus, this approach offers a capability (concept discovery $\to$ localization $\to$ identification $\to$ potential steerability) that neither CLAP nor text prompts are capable of providing.
> >
> > > Could you describe the essentia tags used?
> >
> > We have added a full list as **Appendix D**, in the paper.

---

> > > ### Author Response · Authors · 2025-11-27
> > >
> > > We thank you again for your careful consideration of our work and for acknowledging its merits. We hope to have addressed all your questions in our response, but please let us know if there is anything else we can clarify before the deadline. Otherwise, we hope you could consider increasing your score.

---

### Official Review · Reviewer_1vEC · 2025-11-01

**Soundness:** 2
**Presentation:** 3
**Contribution:** 2
**Rating:** 4
**Confidence:** 4

**Summary:**

This paper extends sparse autoencoder (SAE) methodology from language models to generative music models, specifically MusicGen-Small and Large. The authors train k-sparse autoencoders on residual-stream activations, automatically label discovered features using multimodal LLMs and audio analysis tools (filtered by CLAP alignment), and demonstrate concept steering by intervening on SAE decoder directions. The work reports thousands of interpretable features per configuration, shows that deeper layers in larger models yield more interpretable concepts, and achieves measurable steering success on 15–35% of tested features.

**Strengths:**

To the best of my knowledge, this is the first attempt to apply SAE-based interpretability to audio generation LM; this is a meaningful extension beyond NLP/vision that addresses an underexplored modality. The modular pipeline is well-documented and reproducible. The authors conducted systematic exploration across model scales, layers, sparsity levels, and expansion factors provides useful data on where interpretable structure emerges in music LMs. Table 1 and Fig. 3 offer valuable design guidance. The paper is generally well-written and I find the figures to be informative.

**Weaknesses:**

(1) I notice CLAP serves as both the filter for accepting labels and the metric for evaluating interpretability and steering success. This creates a validation loop where the authors are essentially measuring "does this feature change CLAP scores" rather than "does this feature control meaningful musical concepts"; The human study is too limited to break this circular argument issue.

(2) 15–35% success rate with a single prompt and CLAP-only evaluation is insufficient to claim "robust" controllability. I suggest to add the following: (a) listening studies, (b) diverse prompts, (c) baseline comparisons (e.g., random directions, PCA, supervised probes), (d) negative controls (inactive features, orthogonalized vectors). Per my first point, current results may reflect CLAP bias rather than perceptual control.

(3) Key design choices (θ_min/θ_max thresholds, k values, rarity cutoffs) lack sensitivity analysis. How much do interpretability rates depend on these arbitrary choices? Some statistical report is needed.

(4) Restricting to MusicGen limits this paper's impact. Currently, diffusion/flow-matching based models dominate audio generation; Appendix C acknowledges difficulties but doesn't attempt even a small-scale demonstration (as an example, Stable Audio Open small is small-scale; only 341M parameters). Claims of generality are thus in my opinion unsupported.

(5) Mean-pooling activations across time may obscure temporal dynamics, and there's no analysis of within-track feature stability or polysemantic behavior at different time scales.

**Questions:**

(1) Can you provide human listening tests comparing steered vs. unsteered outputs on perceptual dimensions (genre, timbre, rhythm)? Please compare this with random-direction baselines.

(2) How do interpretability quality metrics (human agreement, not just counts) vary with θ_min, θ_max, k, and EF?

(3) Add negative controls (steering on low-activation features, cross-layer swaps, random directions with matched norm) and report effect sizes with statistical tests.

(4) Can you demonstrate the pipeline on at least one small-scale diffusion-based model to support cross-architecture claims?

(5) How does SAE steering compare to supervised linear probes, activation patching, or MusicGen's own conditioning mechanisms?

---

> ### Author Response · Authors · 2025-11-22
> **[1/3]**
>
> We really appreciate your effort and all the feedback. Thank you for recognizing the novelty, good documentation, reproducibility, scale of experiments, writing, and figures.
>
> > I notice CLAP serves as both the filter for accepting labels and the metric for evaluating interpretability and steering success.
>
> This seems to be a misunderstanding, on two levels:
>
> 1. We do not use CLAP score as a filter, but as a metric for label quality (see Figure 4, where we do not select a threshold, but plot the distribution only). The filtering is done using activation statistics
>
> 2. We point out that this *could be* used as a filter at various different thresholds, which doesn’t change the steering application: it may just change the proportion of features that are successfully steerable (although again, this does not apply in our case)
>
> > 15–35% success rate with a single prompt and CLAP-only evaluation is insufficient to claim "robust" controllability. I suggest to add the following: (a) listening studies, (b) diverse prompts, (c) baseline comparisons (e.g., random directions, PCA, supervised probes), (d) negative controls (inactive features, orthogonalized vectors). Per my first point, current results may reflect CLAP bias rather than perceptual control.
>
> We agree, and this is why we do not claim robust controllability! As we note in the paper:
>
> 1. “the goal of these experiments is to establish the *existence* of steerable concepts”
>
> 2. “This establishes the potential for SAE-driven steering in controllable generation settings, though our primary goal in this work remains feature discovery”
>
> Re: diverse steering prompts, we recognize the potential value of this and it is a challenge more broadly in interpretability research involving steering. For instance, we cited the related work [1] that motivated using a neutral prompt (such as “Simple melody” in our case) in Section 3.7 (Experimental Setup).
>
> Neutral prompts let us (i) cleanly measure a feature's causal influence, (ii) obtain scores that are comparable across features and layers, and (iii) do so in a way that is efficient and independent of variable downstream steering goals. These features make it an appropriate choice for establishing steerability, which is our intent.
>
> We conducted a listening study via Prolific where we recruited 10 participants to each listen to 10 sets of audio clips, where each set contained a baseline, a steered version, a random matched-norm steered version, and a representative of the steering target, asking participants to match the candidates (baseline, randomly steered, and SAE steered). Each participant’s 10 sets were equally sampled from the total of top 50 steerable features. Participants largely selected the SAE-steered audio (66/100, compared to 17 each for baseline and random; $\chi^2 = 48.02,\ p < .0001$). This suggests that the steering effects are clearly perceptible.
>
> [1] Arad, D., Mueller, A., & Belinkov, Y. (2025). SAEs Are Good for Steering--If You Select the Right Features. arXiv preprint arXiv:2505.20063.
>
> [2] Wu, Z., Arora, A., Geiger, A., Wang, Z., Huang, J., Jurafsky, D., ... & Potts, C. (2025). Axbench: Steering llms? even simple baselines outperform sparse autoencoders. arXiv preprint arXiv:2501.17148.
>
> > Key design choices (θ_min/θ_max thresholds, k values, rarity cutoffs) lack sensitivity analysis. How much do interpretability rates depend on these arbitrary choices? Some statistical report is needed.
>
> We appreciate this concern, but emphasize that these are not tuned parameters whose optimality affects any claims. We use them purely as reasonable and literature-driven defaults for filtering viable features. Note that:
>
> 1. The filtering thresholds do not influence the SAE itself. They only determine which features we examine afterwards.
>
> 2. Our results explicitly do not depend on any threshold being optimal. We do not claim the discovered features represent either all possible interpretable features or the "best" such set; we are aiming to establish existence and usefulness rather than exhaustiveness or optimality.
>
> 3. Thresholds are established practice in prior work on SAEs and neural feature sparsity (e.g. using activation prevalence cutoffs to remove degenerate or polysemantic units).
>
> Because the thresholds serve only as a principled mechanism to avoid trivial or pathological features, we believe additional sensitivity sweeps would not meaningfully alter conclusions despite coming at great computational cost.

---

> > ### Author Response · Authors · 2025-11-22
> > **[2/3]**
> >
> > > Restricting to MusicGen limits this paper's impact. Currently, diffusion/flow-matching based models dominate audio generation
> >
> > While diffusion/flow models are increasingly common in audio generation, autoregressive models like MusicGen and AudioGen are still among state-of-the-art open-weight releases. Applying SAEs to diffusion models is an emerging research area that lacks consensus on methodology, unlike the well-established practice of using residual stream SAEs in transformers. For example, Surkov et al. [3] demonstrate SAE application to one-to-few step diffusion models like SDXL Turbo, but current state-of-the-art audio models typically use multi-step diffusion with autoregressive components, so we would need to search across a much larger number of potential hook points, which we believe is computationally infeasible to include within the scope of this study.
> >
> > We have therefore focused on establishing a conceptually clear pipeline that future work can build upon when generalizing mechanistic interpretability methods to such compound architectures.
> >
> > [3] Surkov, Viacheslav, et al. "One-Step is Enough: Sparse Autoencoders for Text-to-Image Diffusion Models." arXiv preprint arXiv:2410.22366 (2024).
> >
> > > Mean-pooling activations across time may obscure temporal dynamics, and there's no analysis of within-track feature stability or polysemantic behavior at different time scales.
> >
> > We agree that time-scale is an important variable. We addressed this by testing a duration-based approach instead (average active duration, rather than average activation strength), but found these to be highly correlated. While it is possible to construct many other measures (e.g. periodicity, moments, etc.) these are difficult to generalize across different features and samples (e.g. tempo changes periodicity), which makes using other such measures much more unprincipled.
> >
> > > Can you provide human listening tests comparing steered vs. unsteered outputs on perceptual dimensions (genre, timbre, rhythm)? Please compare this with random-direction baselines.
> >
> > Thank you for this suggestion, we conducted the listening study and showed the results above.
> >
> > > How do interpretability quality metrics (human agreement, not just counts) vary with θ_min, θ_max, k, and EF?
> >
> > It is unfortunately infeasible to run an adequately powered human listener-based analysis of these questions given the large number of choices and conditions. We note that the thresholds and hyperparameters are set to sensible defaults. We make no claim about the optimality of these values, but at the same time they must be set, and so we defer to common practice in the literature and support with other evidence (CLAP score, feature statistics, and such). This constitutes an achievable lower bound without more hyperparameter tuning.
> >
> > We respectfully request that you reconsider the importance of this analysis to the core contributions: either these values are optimal, or better ones are possible, and we believe that neither case invalidates the obtained results.
> >
> > > Add negative controls (steering on low-activation features, cross-layer swaps, random directions with matched norm) and report effect sizes with statistical tests.
> >
> > In steering experiments, we focus on interpretable, selective features to evaluate their causal impact, and so:
> >
> > 1. Low-activating features are likely spurious which makes it hard to tell if steering fails because steering doesn't work or because the features lack sufficient influence to show it
> >
> > 2. Cross-layer swaps are also unclear as baselines since they confound multiple factors (position, layer specialization, activation magnitude, etc.)
> >
> > We therefore believe your suggestion of random matched-norm directions constitutes a much more reasonable baseline, and have implemented this in the listening study as we noted above, and it is substantially outperformed by the steering in the listening study.
> >
> > We also measured the CLAP score alignment to the target audio for the random-direction vs. steered results, subset to all steerable features. Note that CLAP scores are noisy, and as such we expect a more modest effect here compared to in the listening study. We estimated the CLAP score using a linear mixed-effects regression model, with steering method (random vs. SAE), prompt ("Classical piano", "Jazz saxophone", "Guitar strumming", "Simple melody"), and their interaction. The SAE steering led to an increased CLAP score (marginal effect .011; 95% CI [.004, .019], $p < .001$).
> >
> > Note that a random steering vector does not correspond to a concept and therefore has no associated max-activating examples, and as such no intrinsic target to measure against (we measure against the steering target).

---

> > > ### Author Response · Authors · 2025-11-22
> > > **[3/3]**
> > >
> > > > Can you demonstrate the pipeline on at least one small-scale diffusion-based model to support cross-architecture claims?
> > >
> > > As noted previously, the scale of such an experiment would not be small due to the many potential decisions involved about possible hook points, representations of diffusion activations (e.g. h-space), etc.
> > >
> > > > How does SAE steering compare to supervised linear probes, activation patching, or MusicGen's own conditioning mechanisms?
> > >
> > > It is not feasible to generate a 1:1 comparison between these steering methods, since this would require matching inputs 1:1. For example, it would require: (1) training a probe for each concept we steer, which would require corresponding labeled data, (2) identifying comparable prompt sets for each feature for activation patching, (3) matching prompts and prompt strength against feature strength, etc.).
> > >
> > > The strength of an unsupervised approach such as SAEs is in not requiring these structures to begin with. We make no claims that it would outperform such other methods under a matched comparison; indeed this is not our goal, rather we steer to study the causal properties of the discovered features.
> > >
> > > If there is a commensurable steering intervention comparison you have in mind, as it relates to these different methods, we would welcome the suggestion.

---

> > > > ### Author Response · Authors · 2025-11-27
> > > >
> > > > Thank you again for your reviewing effort, and for appreciating our work. We hope to have addressed all your questions in our response, but please let us know if there is anything else we can clarify before the deadline. Otherwise, we hope you could consider increasing your score.

---

> ### Comment · Reviewer_1vEC · 2025-11-27
>
> Thank you for the detailed response and the additional experiments; I appreciate the effort to address my concerns during the rebuttal period, particularly the inclusion of the random-direction baseline. After reviewing the authors' response, as well as other reviewers' comments and the inclusion of new experiments, I remain concerned that the core methodological limitation, the low 15–35% steering success rate evaluated primarily by CLAP, has not been fully resolved.
>
> While the new listening study is a positive step toward addressing the potential circularity of using CLAP for both labeling and evaluation, a sample size of N=10 participants is statistically too small to serve as a robust validation for a subjective modality like music. (Using CLAP to measure label quality creates similar problem as using CLAP to label.) Without a larger-scale human evaluation, it is difficult to determine if the low steering success rate reflects a limitation of the metric or a fundamental lack of causal mechanism in the discovered features. I do understand, however, that larger-scale human evaluation is costly. It is unfortunate that I believe this study requires such endeavor to verify.
>
> Regarding the comparison to baselines, I respectfully disagree that comparing SAE steering to supervised methods is infeasible. Simpler heuristics, such as mean-difference vectors (subtracting the average embedding of prompt A from prompt B), are standard, low-cost baselines for steering utility; many similar methods exist between the spectrum of the proposed method and this simple constant arithmetic method. By not comparing against such methods, it remains unclear whether the complexity of the SAE approach yields a tangible advantage in control or disentanglement. If human studies are more robust, this point can be let go - as a very impressive subjective listening experiment trumps all; when these two points are compounded, however, the limitation becomes hard to ignore.
>
> If using SAE yields results on par with some simpler, non-sparse method, it could fundamentally mean that the disentanglement of the polysemantic neurons were not successful, and the main discovery will be collapsed to a combination of (1) internal representations of large generative models can be used for semantic alignment, and (2) SAEs can reconstruct internal activation maps, both of which have been previously discovered.
>
> Steerable concepts have to exist in these language models, otherwise they won't function. In my view, the authors have failed to prove that these steerable concepts can also be isolated, which is the main point behind mechanistic interpretability, as we are trying to understand the model in modular, mechanistically sound ways. Combining this with the limitation of using only one model (MusicGen) creates even more doubt to the results' soundness. Thus, while this work is a valuable exploratory step, I maintain my score of 4.

---

> > ### Author Response · Authors · 2025-11-27
> >
> > Thank you for the quick follow-up, we really appreciate the opportunity for discussion. Unfortunately, we think that there are still two key misunderstandings.
> >
> > 1. **On the listening study:**
> > This is a perceptual discrimination task, not a subjective rating task. What matters statistically is the number of independent trials, not the number of unique participants. The study has N = 100 trials, the effect size is large, and the confidence intervals are tight. However, if you believe a different N is required for adequacy, we are happy to run a follow-up at that target. The core point is that the signal is strong and stable, not marginal or noisy.
> >
> > 2. **On the steering comparison:**
> > Our primary contribution is concept discovery, with steering used to test causal influence of discovered features. Comparing this to supervised steering methods misunderstands the role of steering in the paper. Supervised methods cannot perform concept discovery, and therefore fundamentally cannot substitute for or invalidate the unsupervised results. The question we address is whether the discovered units have causal effect, not whether SAEs outperform supervised pipelines on a different objective.
> >
> > Given this, we respectfully ask you to reconsider how much weight you are placing on issues that are not central to the paper’s goals or claims. We believe that the additional evidence we provided directly addresses the methodological questions relevant to those goals.

---

> > > ### Comment · Reviewer_1vEC · 2025-11-28
> > >
> > > Per Reviewer PtMu's new comment regarding the new steering demos; I personally now believe the steering effect is real and achieved, but have some doubts surrounding the "isolated" nature of these activations. Recognizing that interpretability research in language models tend to only include steering effects, I now have the intent to raise my score further.
> > >
> > > The authors raised a point that "Our primary contribution is concept discovery, with steering used to test causal influence of discovered features." My concern was (and still remains to be) that if the discovery process is heavily bound by CLAP, and the steering based on discovered concepts can only be validated by a small group of human listeners (and CLAP), the study's validity is bounded largely by CLAP. In text interpretability research, this point is well-mitigated by how powerful LLMs are at summarization, such that bounded by the capabilities of LLMs are acceptable. I don't believe bounded by CLAP is acceptable. This limitation is inherent to our time, and is possible to be resolved soon as the multimodal language models keep progressing; it is unfortunate that it does place heavy doubt on the paper's findings for me. Some of this doubt was resolved by listening to the steering demos/examples; but I still find the paper's claimed scope to be too large compared to what actually was proven.
> > >
> > > I want to ask for two additional experiments:
> > >
> > > (1) Would it be possible for the authors to conduct additional experiment to show the generation quality is relatively unchanged before/after steering? This could help the "isolated" point. This point would actually be easier to prove if the authors conducted experiments on diffusion/flow-matching models, and can simply do ping-pong sampling [1].
> > >
> > > (2) It would be great if the authors can include experiments on even a very small-scale diffusion/flow-matching model to demonstrate some similar effect. Non-auto-regressive models operating on continuous latent may exhibit very different behavior than auto-regressive language models.
> > >
> > > If both experiments can't be conducted, would it be possible to explicitly acknowledge these limitations in the paper? Especially surrounding (2), since the claim of "Large Generative Music Models" is too broad and unproven; clarifying "autoregressive" would help with the scope.
> > >
> > > I also want to ask the authors to explicitly state how the findings, models and code will be open-sourced in the paper, as I find the steering demos to be highly convincing for me. If all weights and codes (including an example script for demonstrating steering) are open-sourced, it becomes much easier to check these results.
> > >
> > > Provided that both points (acknowledging limitations + clarification on open-source) are resolved, I will raise my score to 6. If more experiments are conducted to resolve rather than simply acknowledge the limitations, I will raise my score to 7 or 8.
> > >
> > > [1] Novack, Zachary, et al. "Fast Text-to-Audio Generation with Adversarial Post-Training." arXiv preprint arXiv:2505.08175 (2025).

---

> > > > ### Author Response · Authors · 2025-11-28
> > > >
> > > > Thank you for your quick response and continued engagement; we really appreciate it.
> > > >
> > > > We understand that it will no longer be possible to update your score, but we wanted to be sure to respond to your points.
> > > >
> > > > We agree re: adding the limitations as you suggested and we plan to do so. Re: the title, we did not mean to imply *all* large generative music models, but we take your point that it could be misinterpreted this way, and will clarify this more strongly in the text. We agree that the discovery performance is bounded by all of the generation (MusicGen), auto-interpretability (Gemini/Essentia-based), and alignment scoring (CLAP) models, and that these models are not as capable as LLMs.
> > > >
> > > > Re: open source, yes we are preparing all code (including steering) and assets for public release. We plan to release it all under a permissive license.
> > > >
> > > > Re: the additional experiments; we acknowledge that, though we believe they are not essential to the paper’s core contribution, they would enhance the contribution and appreciate the suggestions. We will make our best effort to add them to the final version of the paper, if we are able to complete them in time.

---

### Official Review · Reviewer_jYej · 2025-11-01

**Soundness:** 3
**Presentation:** 4
**Contribution:** 4
**Rating:** 8
**Confidence:** 4

**Summary:**

This paper uses SAE to discover interpretable concepts in a pretrained music generation model MusicGen. The process is unsupervised, using pretrained audio taggers/LLM for automatic concept labeling.

**Strengths:**

1. This is the first paper to perform concept discovery with SAE in pretrained music generators, and the results are much better compared to the previous probing works.

2. The design choices in Sec. 3.3 provide very useful insights for future researchers in concept discovery for pretrained audio/music models.

3. The automatic labeling pipeline is promising (with some limitation as stated in weakness) and could be applied to other models (audio generation/understanding models).

In general, this work has the potential to bring significant impact to explanibility in large audio/music models.

**Weaknesses:**

1. In sec. 3.5 automated interpretability, the automatic pipeline might harm the interpretation of some concepts (i.e., chord & keys) since neither gemini nor essentia has such ability.

2. Currently the number of examples are very limited. More examples/case study would be useful in the appendix, including possible failure cases where the automatic labeling pipeline fail to conclude. I.e., more examples where:

(1) A concept could be successfully extracted and named;
(2) A concept could be extracted, but could not be correctly named;
(3) A concept that could not be extracted.

Different types of concepts could also be tested, including genre, instruments, playing techniques, rhythm pattern, groove, sound field, audio quality, mixing techniques, key modes, tempo, velocity, moods etc. Even testing on a portion of them would be very useful.

**Questions:**

1. Line 210: what is a "track" in this context? Does it refer to a single piece of generated music, or a separated stem of a music?
2. Line 397: If later layers encode more interpretable features, why is the numbers in early layers (e.g., L2) larger compared to later layers in Table 1?

---

> ### Author Response · Authors · 2025-11-22
>
> We really appreciate your thorough review, and thank you for acknowledging our novelty, design choices, and promising auto-interpretability pipeline!
>
> > the automatic pipeline might harm the interpretation of some concepts (i.e., chord & keys) since neither gemini nor essentia has such ability.
>
> We agree that the quality of the auto-interpretability model(s) is/are a limiting factor! However it is not clear whether keys and chords are outside the ability of multimodal models such as Gemini. It’s possible to estimate keys using classic signal processing tools, such as those available in Essentia, but we avoided this because many examples in the dataset are not clearly in one key or even tonal.
>
> Nevertheless, we agree that the pipeline is very capable of introducing errors and improving the ability of multimodal models to recognize varied musical concepts would improve these results. On the other hand, as the capabilities of these audio models improve our auto-interpretability results will do so too.
>
> > Currently the number of examples are very limited. More examples/case study would be useful in the appendix, including possible failure cases where the automatic labeling pipeline fail to conclude.
>
> We appreciate the suggestion for including more examples. Our public dashboard (link withheld for anonymity) includes many natural failure cases. It is difficult to highlight failure cases in the Appendix with spectrograms, as most failure cases we observe are cases of *ambiguity* in the label rather than clear mismatch.
>
> For example:
>
> 1. MGL/EF32/k100/L2/F20 is labeled “Synth Arpeggio EDM” by Gemini, but this describes a subset of the max activating examples. As this is an early layer feature, it may be responding to something lower-level and common across different styles that Gemini missed here. This pattern is common to several early-layer features
>
> 2. MGL/EF32/k32/L36/F14435 is labeled “String Instrument Showcase” (description: “String instrument focused music with virtuosic and improvisational elements.”) While this is true, all of the examples are specifically virtuosic solo electric guitar, so the label could have been more specific.
>
> 3. MGL/EF32/k32/L24/F28449 is labeled “Aggrotech”’ (description: “Aggressive electronic music with distorted sounds and repetitive rhythms.”). Though this describes a subset of the max activating examples well, listening to *all* the examples suggest the feature is actually responding to a particular synthesizer texture that happens to be common in this style of music, but also occurs in other settings
>
> > Line 210: what is a "track" in this context? Does it refer to a single piece of generated music, or a separated stem of a music?
>
> A track here is one full clip of music from the dataset, $\approx$10 seconds of audio.
>
> > Line 397: If later layers encode more interpretable features, why is the numbers in early layers (e.g., L2) larger compared to later layers in Table 1?
>
> Thank you for this excellent question. This is a somewhat surprising result, suggesting that there may be a trade-off between feature quantity and feature “quality.” In earlier layers, it may be possible to recover a large number of moderately selective features that are perceptually more ambiguous. In later layers, fewer features may pass this test, but they may be more perceptually stable.
>
> Our new co-activation analyses in **Appendix C** offer another perspective on this issue (see e.g. newly added Figure 12).

---

### Author Response · Authors · 2025-11-30
**Summary for the AC [1/3]**

**Dear AC**

Thank you in advance for your effort in handling this paper. We summarize below the main points and our actions for each reviewer.

We could not have asked for more thoughtful reviewers. We thank the reviewers again for recognizing the novelty and impact of applying SAEs to music generators (**jYej**, **1vEC**, **PtMu**), the clarity, documentation, and breadth of our pipeline and experiments (**jYej**, **1vEC**, **PtMu**), and the broader importance for interpretability in audio and music models (**jYej**, **1vEc**, **eAhf**, **PtMu**).

To summarize our contribution, for context:  We introduce a method for unsupervised discovery of interpretable musical concepts in autoregressive music generation models using sparse autoencoders (SAEs). We focus this study on the residual stream of 2 MusicGen variants. We make this approach scalable and evaluable by developing automated labeling and validation pipelines. Our results reveal both familiar musical concepts and coherent but uncodified patterns lacking clear counterparts in theory or language. As an extension, we show such concepts can be used to steer model generations. Beyond improving model transparency, our work provides an empirical tool for uncovering organizing principles that have eluded traditional methods of analysis and synthesis.

### Summary of new contributions added during rebuttal phase

1. **Appendix C: Large-scale co-activation analysis** showing evidence of reproducible and structured concept discovery
2. **Additional human listening study for steering** (N=100 trials, 10 participants) with random matched-norm steering + unsteered baseline comparisons
3. **Statistical analysis** of CLAP scores comparing SAE vs. random steering, showing improvement from positive steering
4. **Appendix D: Complete Essentia tags list** as requested

---

> ### Author Response · Authors · 2025-11-30
> **Summary for the AC [2/3]**
>
> ### Reviewer **jYej**
>
> - **Automatic interpretability and limitations of Gemini/Essentia for chords/keys.** We agreed that captioning/audio models are a limiting factor and explicitly note that as multimodal models improve, our auto-interpretability pipeline will improve. However, we also pointed out that some of these things (e.g. chord/key recognition) are within the capabilities of e.g. Essentia tools, and possibly also Gemini in some cases.
>
> - **Limited examples**: We discussed our public dashboard (link withheld for anonymity) which includes all 4,000+ features, containing many natural failure cases. We also provided specific examples of ambiguous labeling.
>
> - **Layer-wise interpretability trends**: The reviewer asked why "numbers in early layers (e.g. L2) [are] larger compared to later layers in Table 1" if "later layers encode more interpretable features." We agreed that this is a somewhat surprising result, suggesting that there may be a trade-off between feature quantity and feature 'quality.' In earlier layers, it may be possible to recover a large number of moderately selective features that are perceptually more ambiguous. In later layers, fewer features may pass this test, but they may be more perceptually stable. We also pointed to new co-activation analyses in **Appendix C** for another perspective on this issue (see e.g. newly added Figure 12).
>
> Unfortunately, they didn’t have enough time to respond to our rebuttal before the discussion was terminated early.
>
> ---
>
> ### Reviewer **1vEC**
>
> - **CLAP as both filter and metric**: We pointed out the misunderstanding here: we do not use CLAP score as a filter, but as a metric for label quality (see Figure 4, where we do not select a threshold, but plot the distribution only). The filtering is done using activation statistics.
>
> - **"15–35% success rate with a single prompt and CLAP-only evaluation is insufficient to claim 'robust' controllability."**: We agreed and pointed out that we do not claim robust controllability! As we note in the paper: “the goal of these experiments is to establish the existence of steerable concepts” and “This establishes the potential for SAE-driven steering in controllable generation settings, though our primary goal in this work remains feature discovery.” However, we also **added a new listening study** in response to this point, where participants largely selected the SAE-steered audio (66/100, compared to 17 each for baseline and random; p < 0.001).
>
> - **CLAP score analysis with random baseline**: We carried out this suggestion, measuring the CLAP score alignment to the target audio for the random-direction vs. steered results, subset to all steerable features. The SAE steering led to an increased CLAP score (marginal effect .011; 95% CI [.004, .019], p = 0.002).:
>
> After our initial response, the reviewer noted:
>
> > Thank you for the detailed response and the additional experiments; I appreciate the effort to address my concerns during the rebuttal period
>
> However, they were not yet convinced re: the robustness of the steering evaluation.
>
> After further discussion and listening to steering demos:
> > Per Reviewer PtMu's new comment regarding the new steering demos; I personally now believe the steering effect is real and achieved, but have some doubts surrounding the 'isolated' nature of these activations.
>
> The reviewer then indicated a willingness to increase their score:
>
> > Provided that both points (acknowledging limitations + clarification on open-source) are resolved, I will raise my score to 6. If more experiments are conducted to resolve rather than simply acknowledge the limitations, I will raise my score to 7 or 8.
>
> **Our response**: We agreed to add the limitations as they suggested regarding scope (autoregressive models) and confirmed that we are preparing all code (including steering) and assets for public release. We plan to release it all under a permissive license.
>
> Unfortunately, they will not be able to respond to this last comment because the discussion was terminated early.

---

> > ### Author Response · Authors · 2025-11-30
> > **Summary for the AC [3/3]**
> >
> > ### Reviewer **eAhf**
> >
> > - **Labeling unnamed concepts with multimodal models:** We clarified **this is a misunderstanding**. We use a multimodal model to label features, but it does not necessarily succeed at labeling such unnamed concepts. The labels we suggest for them in the paper are manually specified (as noted in the paper).
> >
> > - **Concerns about steering validity**: We provided listening study results: Participants largely selected the SAE-steered audio (66/100 trials, compared to 17 each for baseline unsteered and random matched-norm steering; p < 0.001). This suggests that the steering effects are clearly perceptible.
> >
> > - **Concerns about temporal aggregation of features across 10 second clips**: We agreed this is a limitation, but not that *some* aggregation cannot be avoided. While it is possible to construct many other measures (e.g. duration as discussed in the paper, periodicity, moments, etc.), these are much less principled (and duration, for instance, is highly correlated with the plain average). We maintain that the mean activation is thus the most sensible baseline approach to implement here.
> >
> > - **Request for co-occurrence analysis**: We conducted a large-scale analysis of co-occurrences detailed in **Appendix C**, which produced several interesting results. We provided examples, including:
> > - Evidence that different SAEs recover the same feature: "Distorted Rock" recovered at MGL layer 36 by three separate SAE configurations
> > - Evidence of semantic hierarchies: "'Wind-Dominated Folk Drone' (Layer 12) → 'Eastern European Folk Wind Music' (Layer 36), MGL EF=32/k=100, 50% overlap between features (5/10 examples)"
> >
> > - **CLAP comparison question** We pointed out that CLAP fundamentally cannot recognize features. Given both an audio clip and a textual label, CLAP can provide an estimated semantic similarity between the two. The problem solved by SAEs is an epistemic one, i.e. unsupervised identification of a coherent internal structure... Thus, this approach offers a capability (concept discovery → localization → identification → potential steerability) that neither CLAP nor text prompts are capable of providing.
> >
> > The reviewer was willing to revise their score as mentioned in their review:
> >
> > > I am a little confused by the premise of this paper, and would appreciate it being explained in the rebuttal (in which case I will be most willing to revise my score).
> >
> > After our rebuttal, the reviewer increased their score (originally **4**). Unfortunately, they didn’t have the chance to respond verbally to our rebuttal before the discussion was terminated early.
> >
> > ---
> >
> > ### Reviewer **PtMu**
> >
> > - **Request for steering demonstration**: We referred to Figure 5 in the existing paper (which contains spectrogram examples, as requested) and added several steering audio examples to our anonymous webpage provided to the reviewers.
> >
> > - **Request for evidence of ‘strong emergence’ vs. CLAP or prompting**: We pointed out that CLAP can evaluate this label and associated examples once they are discovered, but it cannot discover them, which is the main distinction: our pipeline produces perceptually coherent directions in the activation space without requiring supervision. We also provided a case study: a feature for quiet/silence which, despite extensive prompt engineering effort, we were not able to get MusicGen to output via prompting.
> >
> > Despite giving us a strong initial rating of 8, the reviewer noted:
> >
> > > happy to adjust the rating if these issues are resolved.
> >
> > (Though, to be clear, the reviewer’s final response to us before the discussion period concluded did not indicate movement on the score just yet)
> >
> > Reviewer's final assessment after listening to demos:
> >
> > > I listened to the new steering demos and they convincingly address my earlier concern. The key insight—finding meaningful latent directions—is indeed important, and this work is clearly more interpretable than CLAP-style embeddings.
> >
> > The reviewer also expressed thoughts about the distinction between interpretability and controllability, which we agreed with; this paper is very much focused on the former (interpretability), with controllability as supplementary evidence for the potential of concept discovery.

---

### Meta-Review · Area_Chair_KRU6 · 2026-01-07

**Summary:**

This paper proposes using sparse auto-encoders to discover interpretable concepts in autoregressive generative music models. The authors qualitatively demonstrate the value of the discovered concepts, and demonstrate how these could be used to steer model generation.

The reviewers are all positive and supportive of this work, especially after the rebuttal and after having listened to the generated samples. As such, I am recommending this for acceptance, with the important caveat that the authors should make it clear (in the title, abstract, and introduction), that this work is limited to autoregressive models (reviewer 1vEC raised the point that no evaluations are done on diffusion models, for instance).

**Reviewer Concerns:**

Below I highlight the most important concerns raised by reviewers.

## jYej
- W1 (automatic pipeline might harm the interpretation of some concepts (i.e., chord & keys) since neither gemini nor essentia has such ability). This is a somewhat reasonable concern, but also somewhat orthogonal to the paper's contribution. As stated by the authors "as the capabilities of these audio models improve our auto-interpretability results will do so too."
- W2 (the number of examples are very limited... including possible failure cases). The authors adequately responded to this concern, providing descriptions of some informative failure cases encountered.

## 1vEC
- W2 (evaluation is insufficient to claim "robust" controllability). The authors responded well to this concern, providing more human listening studies which yield a significant (i.e. low p-value) result.
- W3 (Key design choices (θ_min/θ_max thresholds, k values, rarity cutoffs) lack sensitivity analysis). The authors provide a reasonable response to this, including "we believe additional sensitivity sweeps would not meaningfully alter conclusions despite coming at great computational cost". Importantly, the authors state that "these are not tuned parameters whose optimality affects any claims".
- W4 (Restricting to MusicGen limits this paper's impact). The authors' response is that it would be computationally prohibitive to expand beyond this. As remarked above, this suggests the authors should make it clear that their contributions are limited to autoregressive models.
- There was a good discussion between the authors and the reviewer, and I believe most (if not all) of the important concerns were properly addressed (modulo the title/abstract/intro change discussed above).

## eAhf
- W2 (The numbers for the steering example don't seem impressive without more information). The authors provided extra listening studies with significant (low p-value) results.
- W3 (For a paper about music, it is very limiting to take the "mean activation" of a feature over 10 seconds.). The authors provided a justification for this which, while not perfect, is reasonable, given the complexity of quantifying musical characteristics.
- Q1 (Do you have any way of measuring “degree to which this is a previously untheorized label”?). The authors respond that they do not, but again, this is part of the complex nature of quantifying musical characteristics.
- Q3 (could you possibly provide examples of co-occurring activating features, and mention if there were any there that were unexpected?).  The authors take this useful suggestion and providing the requested evaluations.
- Q5 (Can you speak as to how your features are different than the features CLAP can recognize, or why your contribution is an improvement over just having access to CLAP?). The authors respond adequately to this: "CLAP fundamentally cannot recognize features. Given both an audio clip and a textual label, CLAP can provide an estimated semantic similarity between the two. The problem solved by SAEs is an epistemic one, i.e. unsupervised identification of a coherent internal structure."

## PtMu
- W1 (Steering demonstration missing.). The authors respond to this adequately, providing more listening studies with significant results (i.e. low p-value), as well as providing steering examples in their website, to which the reviewer replied "I listened to the new steering demos and they convincingly address my earlier concern".
- W2 (Emergent ability needs stronger evidence). The authors respond adequately to this concern: "we note that it’s not necessary that such control dimensions are not mappable to CLAP or other embeddings; the difficulty is finding them".
- W3 (because the automatic labeling step relies on existing music/audio to text models, the performance is somehow bounded by their capability). The authors agree, but as stated previously, this is somewhat orthogonal to the paper's contributions, and "this will improve as
audio/music captioning models continue to improve".

**Reviewer Scores:**

- **jYej:** Currently at an 8, unlikely to increase (and probably not decrease).
- **1vEC:** Currently at 4, the reviewer stated they were willing to increase to 6 or even 8.
- **eAhf:** Currently at 4, but likely to increase given that all major concerns were addressed.
- **PtMu:** Currently at an 8, unlikely to increase (and probably not decrease).

---

### Decision · Program_Chairs · 2026-01-26

Accept (Poster)